# Female mice exhibit resistance to disease progression despite early pathology in a transgenic mouse model inoculated with alpha-synuclein fibrils

Stephanie Tullo [1,2] ✉, Janice Sung Hyun Park [2], Daniel Gallino[2], Megan Park[2], Kristie Mar[2], Vladislav Novikov[3], Rodrigo Sandoval Contreras [3], Raihaan Patel[2,4], Esther del Cid-Pellitero [5], Edward A. Fon [5], Wen Luo [6], Irina Shlaifer [6], Thomas M. Durcan[6], Marco A. M. Prado [3,7,8], Vania F. Prado[3,7,8], Gabriel A. Devenyi[2,9] & M. Mallar Chakravarty [1,2,4,9] ✉

Despite known sex differences in human synucleinopathies such as Parkinson's disease, the impact of sex on alpha-synuclein pathology in mouse models has been largely overlooked. To address this need, we examine sex differences in whole brain signatures of neurodegeneration due to aSyn toxicity in the M83 mouse model using longitudinal magnetic resonance imaging (MRI; T1-weighted; 100 µm³ isotropic voxel; -7, 30, 90 and 120 days post-injection [dpi]; n ≥ 8 mice/group/sex/time point). To initiate aSyn spreading, M83 mice are inoculated with recombinant human aSyn preformed fibrils (Hu-PFF) or phosphate buffered saline in the right striatum. We observe more aggressive neurodegenerative profiles over time for male Hu-PFF-injected mice when examining voxel-wise trajectories. However, at 90 dpi, we observe widespread patterns of neurodegeneration in the female Hu-PFF-injected mice. These differences are not accompanied by any differences in motor symptom onset between the sexes. However, male Hu-PFF-injected mice reached their humane endpoint sooner. These findings suggest that post-motor symptom onset, despite accelerated disease trajectories for male Hu-PFF-injected mice, neurodegeneration may appear sooner in the female Hu-PFF-injected mice (prior to motor symptomatology). These findings suggest that sex-specific synucleinopathy phenotypes urgently need to be considered to improve our understanding of neuroprotective and neurodegenerative mechanisms.

Synucleinopathies, including Parkinson's disease (PD), Dementia with Lewy Bodies (DLB), Multiple systems atrophy (MSA), pure autonomic failure and REM sleep behavior disorder (RBD), are neurodegenerative disorders that share a common feature of aberrant accumulation of alpha-synuclein (aSyn) aggregates in the brain[1–3], which may be underlying the mechanisms of pathogenesis via aSyn spreading[4–6]. To this end, evidence of cell-to-cell propagation has supported the hypothesized prion-like spreading of aSyn that contributes to neurotoxicity and downstream neurodegeneration. Evidence for the prion-like spreading of aSyn emerged from post-mortem analysis of patients' brains that received fetal dopaminergic cell transplantation in the substantia nigra pars compacta[7,8] showing host-to-graft aSyn transfer. Subsequent cellular assays of aSyn cell-to-cell

¹Integrated Program in Neuroscience, McGill University, Montreal, QC, Canada. ²Cerebral Imaging Center, Douglas Research Center, McGill University, Verdun, QC, Canada. ³Robarts Research Institute, Schulich School of Medicine, The University of Western Ontario, London, ON, Canada. ⁴Department of Biological & Biomedical Engineering, McGill University, Montreal, QC, Canada. ⁵Department of Neurology and Neurosurgery, Montreal Neurological Institute, McGill University, Montreal, QC, Canada. ⁶Early Drug Discovery Unit, Montreal Neurological Institute, McGill University, Montreal, QC, Canada. ⁷Department of Physiology and Pharmacology, Schulich School of Medicine, The University of Western Ontario, London, ON, Canada. ⁸Department of Anatomy & Cell Biology, Schulich School of Medicine, The University of Western Ontario, London, ON, Canada. ⁹Department of Psychiatry, McGill University, Montreal, QC, Canada.
✉e-mail: stephanie.tullo@mail.mcgill.ca; mallar.chakravarty@mcgill.ca

transfer[9–14] and animal models of aSyn spreading[15–25] further support the prion-like hypothesis.

Higher prevalence, incidence, increased disease severity and susceptibility in men has been reported in PD[26], MSA[27,28] and DLB[29], and even in PD prodromes, like RBD[30]. The compelling clinical evidence clearly underscores the need to examine the role of sex in disease progression and presentation[31–34]. Yet sex differences in the prion-like spreading of aSyn and the potential downstream consequences are not clearly understood. This is not altogether surprising as sex is often neglected as a biological variable in preclinical research[35–37]. This limitation extends to using the M83 mouse model, harboring an A53T mutation in the human aSyn transgene, that is commonly used to examine the prion-like spreading hypothesis[9,15–25]. Aside from a recent study that used intramuscular injection[23] and a magnetic resonance imaging (MRI) study from our group[24], few others using the M83 mouse line have explicitly explored sex differences. Given the cross-sectional nature of our recent work, it is likely that a longitudinal design may be more sensitive to inter-individual variation that results in sex differences[38,39]. Regardless of sex, most studies using M83 mice characterize aSyn inclusions at a single time point to characterize aSyn spread. Taken together, there is an urgent need to understand sex-specific trajectories using longitudinal study designs to better characterize the sex-specific prion-like spreading of aSyn to better model PD-like pathophysiology.

To this end, we studied the M83 transgenic mice injected with human aSyn preformed fibrils (Hu-PFF) in the right dorsal striatum. We examined sex-specific trajectories of the disease time course using longitudinal structural MRI[40–47]. Next, we derived sex-specific whole brain atrophy patterns at specific focal time points, pre-/peri-motor symptom onset and post-motor symptom onset where the mice neared their humane endpoint. Our study significantly contributes to the expanding body of knowledge, offering insights that could profoundly impact the design and implementation of sex-specific therapeutic interventions for synucleinopathies.

## Materials and methods
### Experimental design
The experimental timeline is shown in Fig. 1 and group numbers in Supplementary Table 1. We examined groups receiving Hu-PFF or PBS in the right dorsal striatum at 7 days prior to the injection (−7 dpi) for a baseline assessment, 30 dpi (earliest signs of aSyn Lewy-body like pathology[18]), 90 dpi (when motor symptomatology onset is typically observed (~ 90–100 dpi)[9,18,22,24,25]), and 120 dpi (when the mice are fully symptomatic and nearing the end stage of the disease. Mice were studied at specified time points ±2 days. Prior to data collection, all mice were weighed. Mice underwent in vivo MRI followed by the administration of behavioral tasks on subsequent days. Motor behaviors were assayed with pole test[48], rotarod[49], and wire hang[15] to assess motor deficits over time. All experiments started with approximately 30 mice per group at −7 days post-injection, with decreasing numbers at each subsequent time point, as mice were diverted at each time point for future terminal experimental methods with a final sample of ~8 mice/sex/group at 120 dpi[41,42,44,47]. We examined sex differences in voxel-wise trajectories of neurodegeneration, neurodegenerative patterns at the two last time points, and phenotypic differences in M83 mice after receiving Hu-PFF inoculation. Finally, we examined whether there were inherent sex differences in the M83 hemizygous mice (not injected), specifically with regards to differences in their normative aSyn expression, as determined by western blotting.

### Animals
We used transgenic hemizygous M83 mice (B6; C3H-Tg[SNCA]83Vle/J) bred in-house (F4-6), expressing one copy of human alpha-synuclein bearing the familial PD-related A53T mutation under the control of the mouse prion protein promoter (TgM83+/−)[50], in addition to the endogenous mouse alpha-synuclein, maintained on a C57BL/C3H background. All mice were bred in-house via the following breeding scheme: TgM83+/− x TgM83+/−. Breeders were supplied by Jax (https://www.jax.org/strain/004479).

The M83 hemizygous mice used here typically present no motor phenotype until 22–28 months of age[50]. aSyn Hu-PFFs were injected in the right dorsal striatum to trigger accumulation of toxic aSyn and accelerate symptom onset[18]. We have complied with all relevant ethical regulations for animal use. All study procedures were performed in accordance with the Canadian Council on Animal Care and approved by the McGill University Animal Care Committee, and the University of Western Ontario (2020-162; 2020-163).

Mice were housed at the Douglas Research Center Animal Facility (McGill University, Montreal, QC, Canada) under standard housing conditions with food and water ad libitum. Mice were typically housed in maximum groups of four, and mice were rarely housed singly unless fighting occurred (only seen in males). Mice were housed with a 12/12-h light/dark cycle, with lights on at 08:00.

### Injection material and stereotaxic injections
Healthy 3 to 4-month-old hemizygous M83 mice were injected with Hu-PFF or phosphate-buffered saline (PBS) in the right dorsal striatum, consistent with previous work modeling known disease epicenters of spreading[9,18,21,24]. Groups were randomly assigned, with an equal number of mice across injection assignment and sex. Post-injection, experimenters were blind to the injection assignment of each mouse during the data acquisition processes and during statistical analysis.

The Hu-PFF were made and characterized by the Early Drug Discovery Unit (EDDU) at the Montreal Neurological Institute using established standard operating procedures (SOPs)[51,52]. Hu-PFF were sonicated and DLS analysis was completed to ensure the average diameter of PFF was <100 nm. PFF were added to 200 mesh copper carbon grid (3520C-FA, SPI Supplies), fixed with 4% PFA for 1 min and stained with 2% acetate uranyl (22400-2, EMS) for 1 min. PFF were characterized using a negative staining protocol[52] and visualized using a transmission electron microscope (Tecnai G2 Spirit Twin 120 kV TEM) coupled to an AMT XR80C CCD Camera, and analyzed with Fiji-ImageJ 1.5 and GraphPad Prism 9 software. Electron microscope characterization of the fibrils in terms of distribution per length can be seen in Supplementary Fig. 1.

The mice were anaesthetized with isoflurane (5% induction, 2% maintenance), given an injection of carprofen (0.1cc/10 grams) and xylocaine was applied on the scalp for pain relief, and the mice were positioned into a stereotaxic platform. Recombinant human alpha-synuclein fibrils (total volume: 2.5 µL; 5 mg/mL; total protein concentration, 12.5 µg per brain) were stereotaxically injected in the right dorsal striatum (co-ordinates: +0.2 mm to +0.2 mm relative to Bregma, +2.0 mm from midline, +2.6 mm beneath the dura). The control condition was M83 mice receiving sterile PBS in the same injection site. Injections were performed using a 5 µL syringe (Hamilton; 33 gauge) at a rate of 0.25 µL per min with the needle in place for about 5 min prior to and after infusion of the inoculum. Different syringes were used for each type of inoculum to prevent any contamination. Post-injection, the mice were placed on a heating pad for recovery before being returned to their home cage.

After inoculation, the mice were monitored weekly for health and neurological signs such as reduced grooming, kyphosis, and/or decreased motor functioning (including reduced ambulation, tail rigidity, paraparesis). The frequency of monitoring increased upon the onset of the symptomatology up until the experimental or humane endpoint. The humane endpoint was defined as having met one of the following criteria: significant weight loss (exceeding 20% of the body weight of a normal mouse), unable to put weight on its hindlegs (falling over when rearing), and/or signs of distress (a rating of "obviously present" as defined by the Canadian Council on Animal Care). At both the experimental or humane endpoint, the mice were anesthetized with ketamine (0.01 ml per 10 grams of mouse's weight) and transcardially perfused with 0.9% (wt/vol) PBS with 0.4% ProHance (gadoteridol, a gadolinium-based MRI contrast agent and 0.1% heparin (an anticoagulant), followed by 10% formalin with 0.4% ProHance) and their brains were collected for future analyses.

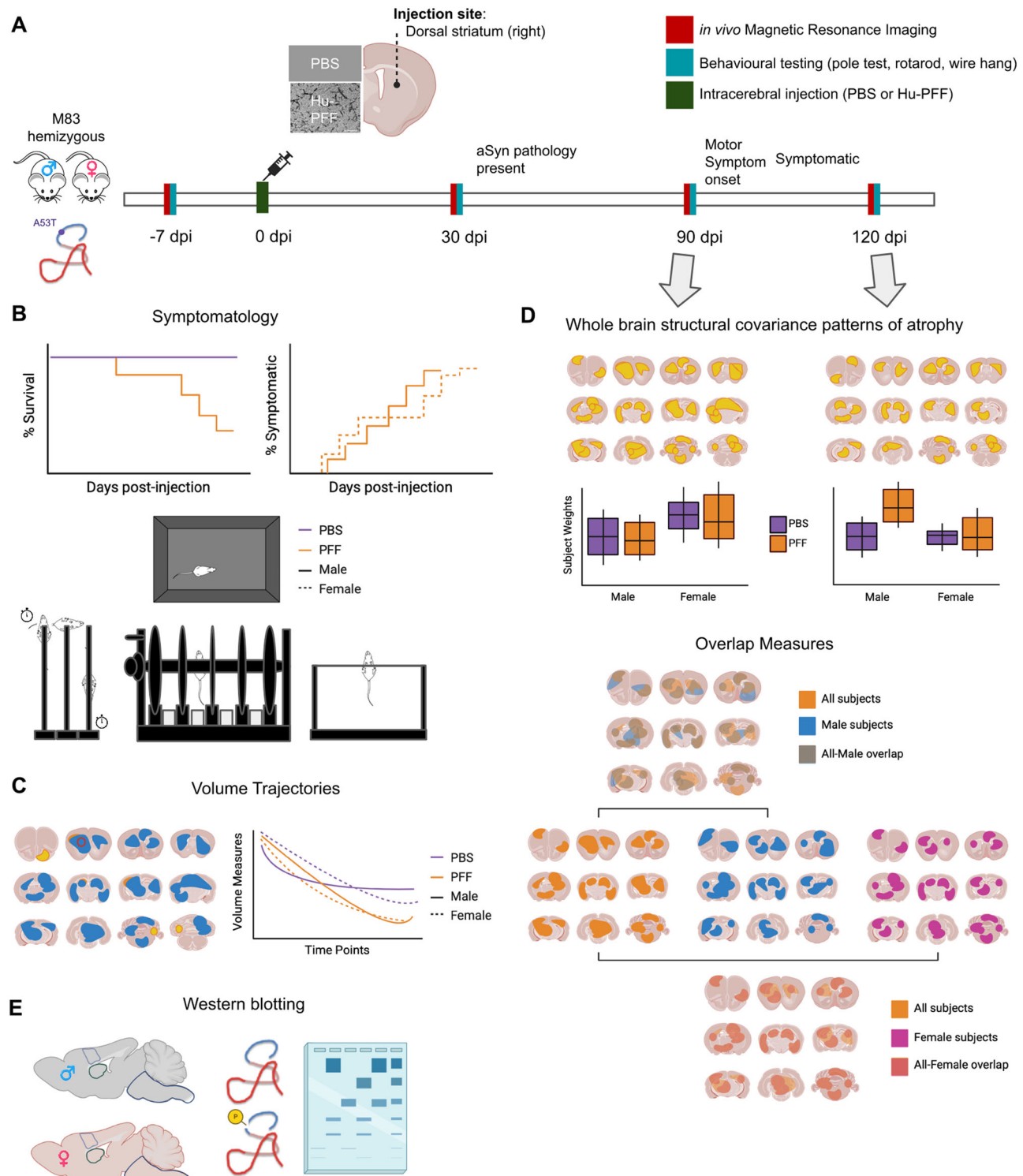

**Fig. 1 | Experimental overview with conceptual figures (not displaying real data).**
**A** Male and female hemizygous M83 mice were injected with either PBS or Hu-PFF; approximately 8 mice per injection group per sex per time point with mice siphoned at each time point. The mice underwent in vivo MRI and motor behavioral testing (pole test, rotarod, and wire hang test) at four time points: −7, 30, 90 (peri-motor symptom onset), and 120 (post-motor symptom onset) days post-injection (dpi). **B** Motor symptomatology was assessed via visual observation of the mice ambulating in their home cage, as well as via their performance on the pole test, rotarod and wire hang test. **C** Univariate analyses were performed to assess group differences in neuroanatomical trajectories across the number of days post-injection.

**D** Subsequently, multivariate analysis (orthogonal projective non-negative matrix factorization) was performed to examine whole brain aSyn Hu-PFF-induced atrophy patterns using voxel-wise measures of volume (Jacobians derived from deformation-based morphometry) at 90 and 120 dpi. Sex differences were evaluated by either examining the injection group by sex differences in NMF subject weights or assessing the overlap between spatial patterning for each sex-specific pattern and the all subjects pattern. **E** Western blotting experiments were performed on non-injected M83 hemizygous mice to examine normative sex differences in these transgenic mice with regards to aSyn expression. Images within figure were created in BioRender. Tullo (2025) https://BioRender.com/k72v615.

## Magnetic resonance imaging (MRI) acquisition

Mice underwent MRI at −7, 30, 90 and 120 dpi, unless they reached a humane endpoint prior. MRI acquisition was performed on 7.0-T Bruker Biospec (70/30 USR; 30-cm inner bore diameter; AVANCE electronics) at the Cerebral Imaging Centre of the Douglas Research Centre (Montreal, QC, Canada). In vivo T1-weighted images (FLASH; Fast Low Angle SHot) were acquired for each subject at each time point (TE/TR of 4.5 ms/20 ms, 100 μm isotropic voxels, 2 averages, scan time = 14 min, flip angle = 20°).

For each time point acquisition, the mice were anaesthetized with isoflurane at 3% induction (with 1% oxygen flow rate) for 3:30 min followed by a bolus intraperitoneal injection of dexmedetomidine (1:240 dilution). The mice remained in the induction chamber until 5:30 min elapsed by which the mice were transferred to the MRI scanner and placed under 1.5% isoflurane with a constant infusion of dexmedetomidine (0.05 mg/kg/h).

## Pre-processing

All brain images were exported as DICOM from the scanner and converted to the MINC (Medical Imaging NetCDF) file format. Image processing was performed using the MINC suite of software tools (http://bic-mni.github.io). At this stage, the images were manually inspected for artifacts (hardware, software, motion artifacts or tissue heterogeneity and foreign bodies) and images with said artifacts were excluded (https://github.com/CoBrALab/documentation/wiki/Mouse-QC-Manual-(Structural)). Injection site was verified for accuracy by examining the resulting edema of the surgery present in the brains at the 30 dpi scan time point. Notably, no remaining edema is present for the subsequent scan time points beyond the 30 dpi time point for any of the mice. No data had to be excluded at this point.

After passing quality control, the images were stripped of their native coordinate system, left-right flipped to compensate for Bruker's incorrect DICOM exporting, denoised using patch-based adaptive non-local means algorithm[53], and affinely registered to an average mouse template (the Dorr–Steadman–Ullman atlas[54–56]) to produce a rough brain mask. Next, a bias field correction was performed, and intensity inhomogeneity was corrected using N4ITK[57] at a minimum spline distance of 5 mm.

## Image processing: deformation-based morphometry

After scanning all images underwent pre-processing and quality control prior to being used as input in the deformation morphometry pipeline. Here, subjects are aligned with a series of linear and nonlinear registration steps with iterative template refinement[58] to generate a study-specific template enabling voxel correspondence across every subject timepoint. This method has been previously used by our group to examine voxel-wise volumetric change over time[41,42,44,47].

Given the longitudinal nature of the data, a two-level deformation-based morphometry technique (antsMultivariateTemplateConstruction2.sh; https://github.com/CoBrALab/twolevel_ants_dbm) was used to perform group-wise registration of all MRI data for a single subject to first generate a subject-wise template (first level), followed by the registration of all the subject-wise templates to create a final group-wise template (second level) to enable statistical analysis in a common space. The final deformations fields represent the minimum deformation required, at a voxel-level, to map each subject time point image to the group template. The Jacobian determinants were log-transformed then blurred with Gaussian smoothing using ~0.085 mm full width at half maximum kernel to better conform to normative distribution assumptions for statistical testing[59].

Voxel-wise volume measures were derived from two types of Jacobian determinants, depending on the type of analysis. For longitudinal analysis, "relative" Jacobian determinants (exclusively modeling the nonlinear transformations of the deformations) were used to measure local anatomical differences, whereas "absolute" Jacobian determinants of the deformations were used to measure all anatomical differences (including the residual global linear transformations attributable to differences in total brain size for example along with the nonlinear transformations) for cross-sectional analyses (specifically, for the whole brain structural covariance analyses

below). In both cases, these Jacobians represent expansions (positive values) or contractions (negative values).

The outputs were inspected to assess quality control of the registration by visually assessing the resultant images of the registrations for a proper orientation, size, and sensible group average, prior to performing the analyses (http://github.com/CoBrALab/documentation/wiki/Mouse-QC-Manual-(Structural)). No data had to be excluded at this point.

## Motor behavior tasks

After scanning at each of the four MRI timepoints (−7, 30, 90, 120 dpi), the mice underwent a behavioral test each day for 3 days (>24 h post-scan) to quantitatively measure motor symptomatology; namely: pole test, rotarod and wire hang test. Tests were conducted during the light phase, between 8 a.m. and 8 p.m. The mice were habituated to the testing room for 1 h prior to administering the behavioral test.

The pole test was administered to examine motor agility and fine motor movements[48]. Each mouse was placed head-upward on the top of a vertical rough-surfaced pole (wooden dowel covered in surgical tape; diameter 8 mm; height 55 cm). The time required for the mouse to descend to the floor was recorded with a maximum duration of 2 min (120 s). Each subject completed three trials with a 30-min rest between each trial, following the protocol detailed by Matsuura et al.[48]. Notably, two scores were recorded: (1) whether the task was performed successfully or was a failed attempt, and (2) the time taken to descend. If a mouse was not able to turn downward and remained at the top of the pole, the trail was marked as a failure and the maximum time allotted was noted (120 s). In cases where the mouse fell part of the way down but subsequently descended the rest, the behavior was scored as a failure however the time it took to reach the floor was still noted. However, if the mouse fell for more than half the length of the pole, the behavior was scored as a failure and the maximum time allotted was noted. For cases where a mouse fell immediately after placement at the top of the pole, a failed attempt, the trail was repeated.

To investigate motor coordination and balance, mice were tested on the Rotarod apparatus[49]. Mice were placed on the Rotarod (San Diego Instruments; San Diego, CA, USA) and rotation was accelerated linearly from 4 to 25 revolutions per minute (rpm), increasing rpm every 15 s. Time spent walking on top of the rod before falling off the rod or hanging on and riding completely around the rod was recorded. Mice were given three trials with a minimum 30-min inter-trial rest interval.

Neuromuscular strength was tested with the wire hang test. The mouse was placed on a wire by waving it gently so that it gripped the wire and then inverted. Latency to fall over the course of three trials was recorded with a 3-min cut-off time, with a 30-min inter-trial rest period[15]. Similar to the pole test, performance on the wire hang was examined in terms of latency and failure/success rates. If the mouse holds on for more than 3 min, it receives the maximum allotted time (150 s) and a successful score. For this task, there are three possible outcomes: (1) the mouse holds on for more than 3 min, (2) the mouse holds on for less than 3 min and receives a failure score with the duration recorded, or in the most severe case, (3) the mouse is unable to hold on for at least 10 s. In the latter case, three attempts were allowed during the trial before a failure score and a duration of less than 10 s was recorded (of highest duration of the failed attempts).

## Western blotting

Cortical, striatal, and brainstem tissue from 6–7 months old non-injected M83 hemizygous mice ($n = 8$ mice; 4 males and 4 females) was homogenized to obtain radioimmunoprecipitation assay (RIPA) soluble and insoluble fractions to examine differences in aSyn expression between the sexes. The following primary antibodies were used: phospho S129 (1:1000, Cat# ab51253, Abcam, RRID:AB_869973), anti-human a-Syn (1:1000, Cat# ab27766, Abcam, RRID:AB_727020), anti-alpha synuclein (1:1000, Cat# 610787, BD Biosciences, RRID: AB_398108), anti-actin HRP (1:25,000, Cat#A3854, Sigma-Aldrich, RRID:AB_262011). Densitometry was processed and analyzed using Bio-Rad ImageLab software.

The brain tissue was rapidly dissected, flash frozen using dry ice, and stored at −80 °C. Cortical, striatal, and brainstem tissue was homogenized using ice-cold RIPA buffer (50 mM Tris pH 8.0, 150 mM NaCl, 5 mM EDTA, 0.1% SDS, 0.5% Sodium Deoxycholate, 1% Triton-X 100) with phosphatase inhibitors (1 mM NaF and 0.1 mM Na3VO4) and protease inhibitor cocktail (1:100, Catalog#539,134-1SET, Calbiochem). Specifically, a handheld homogenizer with a pestle was used to break down the tissue, followed by three rounds of 7-s sonication at 4 °C, and finally centrifugation for 20 min at $17,000 \times g$. Supernatant containing RIPA-soluble homogenates was collected, aliquoted, and stored at −80 °C. The remaining pellet was washed with ice-cold PBS, suspended in 4 M Urea and 2% SDS, vortexed for 10 s at max speed, sonicated, and centrifuged.

Supernatant concentration was determined using ThermoFisher BCA protein assay kit (Cat# 23227), samples were prepared with 4x sampling buffer (40% Glycerol, 250 mM Tris-HCl, 8% SDS, 4% β-mercaptoethanol, 6 mM Bromophenol blue), boiled at 95 °C for 5 min, and 13 ug of protein was loaded into precast 4–12% Bis-Tris gradient gels (Thermo Fisher). Protein was transferred onto 0.2 μm PVDF membranes, fixed in 0.4% PFA for 30 min at room temperature, stained with 0.1% Amido black, and blocked with either 5% milk or 5% BSA in 1x TBS-T. Membranes were incubated in the following primary antibodies: phospho S129 (1:1000, Cat# ab51253, Abcam, RRID:AB_869973), anti-human a-Syn (1:1000, Cat# ab27766, Abcam, RRID:AB_727020), anti-alpha synuclein (1:1000, Cat# 610787, BD Biosciences, RRID: AB_398108), anti-actin HRP (1:25,000, Cat#A3854, Sigma-Aldrich, RRID:AB_262011). Sheep anti-mouse HRP (1:5000, Cat#SAB3701095, Sigma-Aldrich, RRID: N/A), and goat anti-rabbit HRP (1:10,000, Cat#170–6515, Bio-Rad, RRID: AB_11125142) were used as secondary antibodies. Membranes were exposed using chemiluminescence and imaged the ChemiDoc MP Imaging System. Densitometry was processed and analyzed using Bio-Rad ImageLab software.

## Statistics and reproducibility

Disease progression (in terms of survival) and motor symptom onset were examined using Cox Proportional Hazard modeling[60]. This cumulative incidence model is considered superior to the *t*-test or ANOVA given the nature of the data where two measures can be used to denote the behavior of the mouse. For the survival and motor symptom onset analysis, we examined (1) the average dpi overt motor symptomatology was first observed in ambulating mice or humane endpoint dpi as well as (2) the proportion of the mice at that dpi. For the pole test and wire hang test, Cox modeling is best suited given that these tests are conducted with max latency times (3 min success cut-off for wire hang test, and 2-min failure cut-off for pole test)[41].

For these motor tasks, the Cox Proportional Hazard models test the effect of the injection groups and sex interaction to explain the rates of successfully completing each motor task (independently) over time, along with trial number and weight as covariates. This model allows for non-parametrically distributed latency times, the unbiased inclusion of failures (trials where mice cannot perform the task due to motor impairment and thus counted as a max time value (for pole test) or obtain a zero value (for wire hang)), and the ability to include covariates and test for their explanatory contributions[41,61].

We also examined differences in average dpi overt motor symptomatology was first observed in ambulating mice using general linear model (average dpi ~ injection_group*sex). For longitudinal measures such as weight and rotarod performance (averaged across trials at each time point), linear mixed effect models were used to assess injection group by sex differences across the time points (~ injection_group*sex*time point + (1| subject_ID)). For pole test and wire hang tests, performance was examined with regards to latency and failure/success rate using a Cox Proportional Hazard model[60] as these two tasks have strict time cut-offs to determine the success/fail rate. Distribution of data, highlighting a ceiling and floor effect for wire hang and pole test respectively, is available in the Supplementary Figs. 2 and 3. Bonferroni correction was performed to account for multiple comparisons ($p = 0.05/8$; 2 motor tests and 4 time points). Statistical analyses were carried out using R software (3.5.0) with the RMINC package

(https://github.com/Mouse-Imaging-Centre/RMINC) for voxel-wise analysis, and the survival package (https://cran.r-project.org/web/packages/survival/index.html) for the cox modeling. NO data points were excluded for any statistical analysis.

Next, we performed univariate analyses to assess group-level voxel-wise differences in neuroanatomical change using the relative Jacobian determinants (exclusively modeling the nonlinear transformations, representing volume differences, that were log transformed and blurred using Gaussian smoothing) generated from DBM using linear mixed effects models at each voxel to examine volume differences between the injection groups and sex of the mice over time:

Relative volume at each voxel ~ injection_group*sex*time point + (1| subject_ID)

Statistical findings were corrected with False Discovery Rate (FDR)[62]. Statistical analyses were carried out using R software (3.5.0) and the RMINC package (https://wiki.phenogenomics.ca/display/MICePub/RMINC).

## Orthogonal projective non-negative matrix factorization

We previously demonstrated that modeling MRI-derived atrophy in M83 mice with striatal inoculation using methods that capture spatial covariance patterns[24] resulted in substantial homology to human patients with PD[63–65]. In our previous work we used a multivariate technique known as orthogonal projective non-negative matrix factorization (OPNMF)[66–69] (https://github.com/CoBrALab/cobra-nmf).

OPNMF decomposes an input matrix, dimensions $m \times n$, where $m$ is the number of voxels and $n$ the number of subjects into two matrices. Here, the input matrix was composed of the inverted absolute Jacobian determinants $z$-scored at each voxel for each subject, loaded as columns. The first output matrix is the component matrix, dimensions $m \times k$, where $k$ represents the number of components (i.e. spatial patterns of covariance). This output matrix is composed of the component scores for each voxel, which describes the groupings of voxels sharing a covariance pattern. With these voxel-wise component weights, the spatial pattern of voxel weights for each component can be plotted onto the mouse brain template generated by DBM. Moreover, with the orthogonality constraint of OPNMF, this variant of NMF allows for the output spatial components to be non-overlapping, and that each output component represents a distinct pattern of voxels. Thus, a specific voxel can only be part of one component, however the voxels within a region can be included in more than one component. The second output matrix is the weight matrix, dimensions $k \times n$, which contains the weightings of each subject for each component, thereby describing how each subject loads onto each pattern. The component and weight matrices are constructed such that their multiplication reconstructs the input data as best as possible by minimizing the reconstruction error between the original input and the reconstructed input.

To select the optimal number of components ($k$) to analyze, we emulated the stability analyses performed in Patel et al.[66] performed at various granularities (from $k = 2$ to $k = 20$; at intervals of 2). Two measures are commonly examined when assessing stability and determining $k$. First, accuracy is measured by observing reconstruction error of a decomposition, defined as the Frobenius norm of the element-wise difference between the original and reconstructed inputs. We plot the gradient in reconstruction error, enabling quantification of the gain in accuracy provided by increasing the number of components from one granularity to the next. Second, stability of a decomposition is measured by assessing the similarity of output spatial components across varying splits of subjects. To track stability, OPNMF is performed on various subsets of subjects and the spatial similarity of the resulting outputs describes how consistent results are across the sample. Here, 5 splits were performed at each granularity.

A high stability value with a low gradient in reconstruction error (i.e. a smaller gain in accuracy by increasing the granularity) is optimal. Here, we define optimal as a balance between high stability (indicating the spatial patterns are consistent across subjects), low reconstruction gradient (indicating that the increase in $k$/added complexity does not confer a large increase in reconstruction accuracy), and low $k$ (fewer components

represents a more compact deconstruction and less chance of overfitting) Notably, the gradient in reconstruction error increases as the granularity increases, however at higher granularities there is a greater propensity of overfitting. See Supplementary Fig. 4 for plots of stability and gradient in reconstruction error for each OPNMF.

Here, $k = 6$ was selected for the OPNMF performed at both time points. $k$ was selected based on the stability of each decomposition (i.e. the similarity of output spatial components across varying splits of subjects) and the accuracy as measured by the gradient in reconstruction error (Frobenius norm of the element-wise difference between the original and reconstructed inputs), as detailed in Patel et al.[66].

Importantly, this methodology contains no information (injection group nor sex) regarding the mice under investigation. Thus, for each spatial pattern (each component), general linear models were used to examine the sex by injection group interaction in the subject weights post-hoc for each component to identify sex differences.

Beyond the examination of the OPNMF subject weights to examine inter-individual variation, since the patterns that are elucidated are threshold free and are sparse matrices, they can be used to evaluate the overlap between the pattern that best differentiates the Hu-PFF- and saline-injected groups as well as the male- and female-specific aSyn-mediated neurodegeneration to better understand sex-specific overlap with the group pattern. We measured the overlap between sex-specific and whole group (all subjects) binarized maps to assess pattern shape and extent using the Dice similarity coefficient (also referred to as Dice's Kappa) ($\kappa$) metric. This is similar to recent work from our group used to quantify network shape in rodent resting state functional MRI[70].

The Dice's Kappa overlap metric score is: $\kappa = 2a/(2a + b + c)$; where $a$ is the number of voxels common to comparative spatial patterns, and $b + c$ is the sum of the voxels uniquely identified in each pattern. A higher $\kappa$ value denotes a higher degree of overlap, where a score of 0 represents no overlap and a value of 1 represents perfect overlap[71].

### Reporting summary
Further information on research design is available in the Nature Portfolio Reporting Summary linked to this article.

## Results
### Assessing disease progression and motor symptom profile
As expected, we observed lower survival rates for Hu-PFF-injected mice, however, we observed that male Hu-PFF-injected mice reached their humane endpoint prior to the end of the experiment (120 dpi) ($n = 84$). This significant injection group by sex interaction ($p = 0.000999$; Cohen's $d = -3.42$) suggests a significant physiological difference in response to the Hu-PFF injection in males and females (Fig. 2A). Male Hu-PFF mice reached their humane endpoint after an average of 23 days of overt motor symptomatology.

Given the differences in sex-specific survival rate, we assessed differences in the length of time required for the mice to be considered symptomatic; however no sex-specific differences were observed (Fig. 2B) ($n = 84$). Similarly, no significant sex differences in the number of days after which motor symptoms were first observed between the two injection groups were observed (Fig. 2C). Contrary to our survival analysis, our findings do not suggest a sex difference in the onset and rate of onset of overt motor symptomatology.

Beyond observational motor symptoms, we examined motor behavior and disease progression via weight tracking and motor performance. For the M83 mouse model, initial phenotypic changes include neglect of grooming, weight loss, and reduced ambulation[50]. We observed a significant inverted U-shaped trajectory for Hu-PFF-injected mice, compared to the consistent weight increase in PBS-injected mice ($n = 329$; $p = 0.00108$; $d = -3.31$). In the M83 mice injected with Hu-PFF, weight peaks at ~90 dpi then declines steadily thereafter. However, no sex-specific differences were observed when examining the triple interaction between group, sex, and time since injection (Fig. 2D).

At each timepoint, performance on the rotarod, wire hang and pole test were assessed. Average rotarod performance across time showed no significant differences for the sex by injection group by time interaction (Fig. 2E); nor were there group by sex differences in performance at 90 and 120 dpi ($n = 256$). Performance was evaluated cross-sectionally at each timepoint for both wire hang and pole tests in terms of failure/success rates, and performance duration. Sex-specific injection group differences are displayed in Supplementary Fig. 5. Hu-PFF-injected mice had significantly worse performance (higher failure rates and shorter hang duration) at both 90 dpi ($n = 186$; $p = 0.0132$; hazard ratio [HR] = 1.74) and 120 dpi ($n = 93$; $p = 0.00032$; HR = 4.03) on the wire hang test compared to PBS (Fig. 2F, G). Similarly, for the pole test, these mice had significantly worse performance (lower proportion of successfully passing the task and longer time to descend the pole) at 90 dpi ($n = 189$; $p = 0.0304$; HR = 0.603) and 120 dpi ($n = 87$; $p = 0.017$; HR = 0.229) (Fig. 2H, I).

### Longitudinal volumetric analysis
We observed significant differences in the rate of volume change as time progressed for Hu-PFF in comparison to PBS mice (Fig. 3A). Similar to our previous findings, these significant differences were mainly observed in the key structures implicated in synucleinopathies[24], as well as with known connections to and from the injection site (right caudoputamen); namely, in voxels in the contralateral striatum, bilateral somatomotor cortices, periaqueductal gray, as well as highly connected regions such as voxels within the bilateral thalami, all surviving 1% False Discovery Rate (FDR)[62].

There were significant sex-specific differences in volumetric change over time. We observed a significant three-way interaction of the injection group by sex by time, with the fastest rates of neurodegeneration occurring in male Hu-PFF-injected mice ($n = 271$). This accelerated rate for the males was localized in voxels within the injection site (right striatum) (Fig. 3G, K), the ipsilateral substantia nigra (Fig. 3F, J), bilateral motor cortices (primary and secondary motor areas) (Fig. 3H, L), and midbrain areas such as the midbrain reticular nucleus (Fig. 3E, I). We explicitly examined injection group differences over time for each sex separately to better assess sex-specific influence of Hu-PFF injections (Fig. 3B–D). Qualitatively, we observed greater differences between Hu-PFF- and PBS-injected mice for the male mice ($n = 139$) compared to the same comparison in female mice ($n = 132$) (Fig. 3B, C). Such differences between each sex-specific comparison can be best seen in the caudoputamen both ipsilateral and contralateral to the injection site, and is particularly striking for more posterior areas such as midbrain and brainstem areas. Accordingly, we examined sex differences for the Hu-PFF-injected mice over time ($n = 138$) (Fig. 3D), and we observed steeper rates of decline for male mice, particularly in regions such as the contralateral caudoputamen (Fig. 3N).

### Whole brain structural covariance patterns of atrophy
In addition to examining volumetric trajectories of decline to toxicity derived from Hu-PFF injection, we sought to elucidate if this decline was occurring in a network-like pattern. To examine this, we used a multivariate statistical methodology similar to methods that have previously been used in clinical PD[64] and that are consistent with our previous work[24]. The technique we have favored, namely the orthogonal projective non-negative matrix factorization (OPNMF)[66–69], uses the voxel-wise measures of volumetric differences as input and outputs. It provides voxel-level spatial components that covary together and subject-specific weights related to how much each subject loads onto a spatial pattern.

Similar to the analysis performed in Tullo et al.[24], at 90 dpi, we examined the spatial patterns for $k = 6$ components (as determined by stability analysis; Supplementary Fig. 4). The focus of this analysis was to examine data-driven patterns of PFF-induced atrophy and their sex differences ($n = 87$). General linear models for each of the six components revealed no significance with sex-by-injection group interaction that predicted subject-wise loading onto the spatial pattern (Fig. 4A). One of the six components statistically separates the Hu-PFF- from PBS-injected groups (component 1; $p = 0.0261$; $d = 2.266$) (Fig. 4B). The spatial covariance

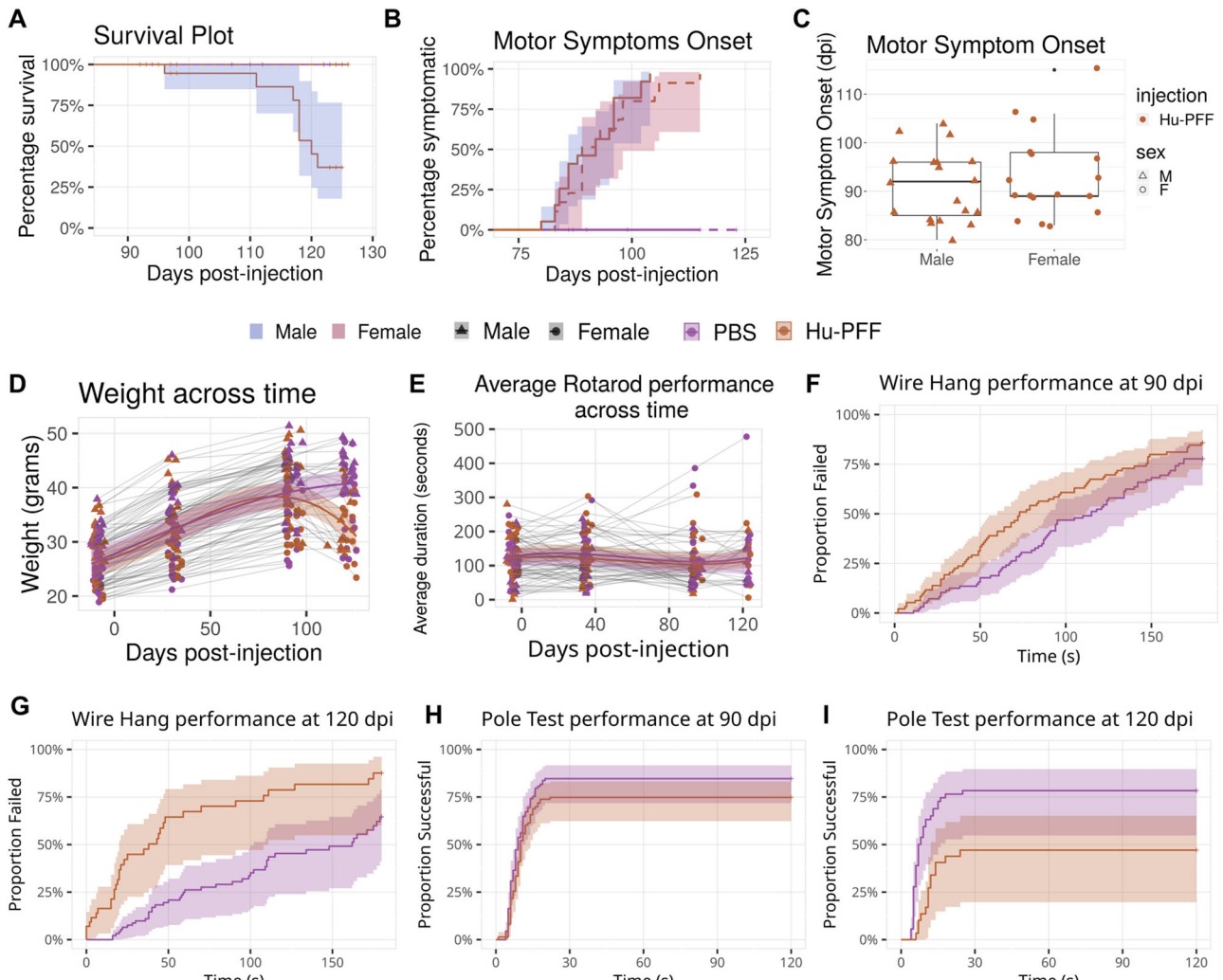

**Fig. 2 | Sex-focused examination of disease progression and motor symptomatology. A** Survival plot for M83 hemizygous mice. Percent survival of the mice plotted across the days post-injection (dpi) with the experimental endpoint being 130 dpi. Almost 30% of the male Hu-PFF-injected mice succumbed to their symptoms/disease progression a couple days prior to the 120 dpi time point. Group differences in survival rate were examined using general linear models, and a significant group by sex interaction was observed whereby male Hu-PFF-injected mice had lower survival rates ($n = 84$ mice; $p = 0.000999$). Of the mice that did not survive within the time frame examined (<130 dpi), the mice sustained motor symptomatology for on average 23 days. Shading represents a 95% confidence interval. **B** Percentage of mice showing motor symptoms across the number of days post-injection. No significant differences between the sexes were observed with regards to the percentage of mice being symptomatic ($n = 84$ mice). Shading represents a 95% confidence interval. **C** With regards to motor symptom onset, in terms of days post-injection, no significant differences were observed between the sexes ($n = 84$). The error bars indicate the data spread up to 1.5 times the interquartile range. **D** Weight trend across disease progression. Significant inverted U-shaped trajectory for Hu-PFF-injected mice, with weight loss as of 90 dpi (compared to PBS-injected mice) ($n = 329$ mice; $p = 0.0056$), regardless of sex. Shading represents ±1 standard error of the mean. **E** Average rotarod performance across time showed no significant differences between injection groups and sex ($n = 256$ mice). Shading represents ±1 standard error of the mean. **F–I** When interpreting the plot, the $y$-axis (Proportion Failed/Successful) shows the percentage of mice (or subjects) that failed to last more than 3 min on the wire hang test or successfully completed the pole test. The closer the line is to 100%, the more successful/unsuccessful the subjects were. The $x$-axis (Time in seconds) indicates the time taken to successfully complete the pole test/the amount of time they lasted on the wire under the 3-min successful cut-off. As time increases, the plot tracks the proportion of subjects who were successful/failed by that time. Next, the lines show the cumulative success/failure rates over time for the two injection groups. Shading represents a 95% confidence interval. **F**, **G** Wire-hang performance at 90 (left) and 120 (right) dpi. Significant difference in the proportion of mice that failed (<3 min) between injections groups (90 dpi: $n = 186$ mice; $p = 0.0132$; 120 dpi: $n = 93$ mice; $p = 0.00032$), with higher rates of failure for the Hu-PFF-injected mice (red dashed line); no sex difference observed. **H**, **I** Pole Test performance at 90 (left) and 120 (right) dpi. Hu-PFF-injected mice had lower proportions of mice successfully passing the test, and took significantly longer to descend the pole compared to their saline injected counterparts (90 dpi: $n = 189$ mice; $p = 0.0304$; 120 dpi: $n = 87$ mice; $p = 0.017$). See Supplementary Fig. 5 for visualization of non-significant injection groups by sex differences for the weight and rotarod analysis, and sex-specific injection group effects for the wire hang and pole test at 90 and 120 dpi. Purple color denotes PBS-injected mice and orange color denotes Hu-PFF injected mice. Line type was used to denote each of the sexes: solid line (with blue shading) for male and dashed line (with red shading) for female mice, except when no sex differences are displayed solid lines then denote both sexes grouped together. Data point shapes also denote the sex of the mice: triangle for males and round for females.

pattern of this component (component 1) consisted of voxels in regions with known connections to the injection site, such as voxels within striatal-pallidal-midbrain areas, as well as strong cortical and thalamic involvement (Fig. 4A). All six components are detailed in Supplementary Fig. 6. For this component (component 1), we observed a significant effect of sex ($p = 2.62e$ −07; $d = 5.621$), with higher subject weights for all females (Fig. 4C), suggesting that female mice (regardless of injection group) had stronger associations with this spatial covariance pattern than the males. We used these findings to support further examination of sex-specific structural covariance patterns.

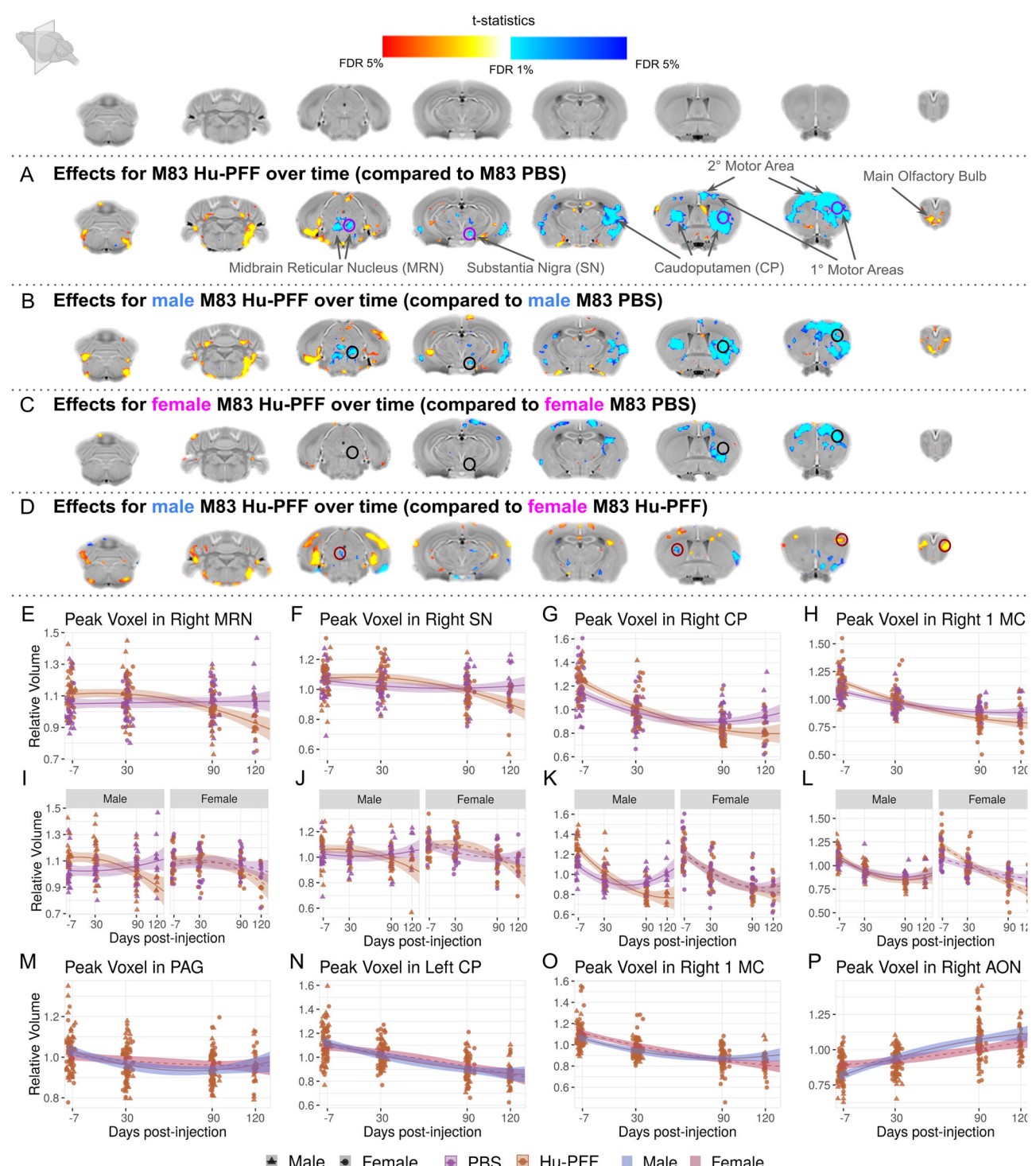

For each sex-specific OPNMF ($n = 44$ males; $n = 43$ females), we similarly observed one of the six components with significantly higher subject weights for the Hu-PFF-injected mice, compared to their saline counterparts (component 5 for the male only analysis ($p = 0.00182$; $d = 0.178$) and component 1 for the female only ($p = 0.0113$; $d = 2.652$) (see Supplementary Figs. 7 (male subjects) and 8 (female subjects) for all 6 components in each sex-specific analysis).

Both the male- and female-specific Hu-PFF-induced patterns (component 5 and component 1 respectively) consisted of voxels within the same regions mentioned above for the all subjects OPNMF analysis; which included: voxels in the basal ganglia and thalamus, anterior to posterior

dorsal cortical areas, and in the midbrain and pons (Fig. 4D). To further examine the degree of similarity between each sex-specific OPNMF and the original all subject OPNMF, Dice-kappa overlap scores were used to examine the degree of overlap between each sex-specific and all-subjects patterns as a proxy for sex-specific contributions. Our analysis revealed a higher degree of overlap between the females and the all subjects pattern ($\kappa = 0.62$) comparatively to the male only overlap ($\kappa = 0.46$) (Fig. 4D). These findings suggest that, in contrast with our univariate analysis that reveals more aggressive localized neurodegeneration in male mice, there is a more spatially widespread neurodegeneration pattern for the female mice compared to their male counterparts associated with the PFF injection.

**Fig. 3 | Sex differences in Hu-PFF-induced brain atrophy examined using voxel-wise volumetric trajectories over time.** Coronal slices of the mouse brain (from posterior to anterior) with *t*-statistical map overlay; demonstrating **A** the effects of injection over time in M83 hemizygous mice (*n* = 271 mice), **B** the effects of injection over time in male mice (*n* = 139 mice), **C** the effects of injection over time in female mice (*n* = 132 mice), **D** the effect of sex over time for Hu-PFF-injected mice (*n* = 138 mice). Color map describes the direction of the *t*-statistics; cooler colors denoting negative values; most commonly corresponding to volume decline and warmer colors denoting positive values, corresponding to volume increases over time for the group of interest indicated. **E–P** Plot of relative volume change (mm³) over the four time points (−7, 30, 90 and 120 days post-injection). Shading represents ±1 standard error of the mean. We chose to highlight voxels within regions where group differences over time were observed (as observed in (**A**)) then examined if these effects would be observed in each sex-specific analysis (as observed in **B** for the males and **C** for the females); peak voxel in (**E**, **I**) the right midbrain reticular nucleus (MRN),

**F**, **J** right substantia nigra (SN), **F** the injection site (right caudoputamen (CP)), **H**, **L** right primary motor area (1 MC). Our observations for sex differences are supported by this style of examination. In these regions we observed significant group differences over time for males but not females. Purple line for PBS-injected mice, orange line for Hu-PFF-injected mice, solid line and triangle points for male and dashed line and circular points for female mice. Separately, plots **M–P** are displaying sex differences in voxel-wise trajectories for the Hu-PFF mice only, comparing the trajectories between male and female mice. Here, we selected peak voxels whereby we observed significant differences in trajectories; regions including the **M** PAG, **N** contralateral caudoputamen, **O** ipsilateral primary motor cortex and **P** anterior olfactory nucleus; orange points and line describe these Hu-PFF injected mice, where solid line, triangle points, and blue shading describes the male while the dashed line, circular points and red shading describe the trajectory for the female mice. Overall, volumetric decline was observed for Hu-PFF injected mice (compared to PBS-injected mice), with steepest rate of decline for male Hu-PFF-injected mice.

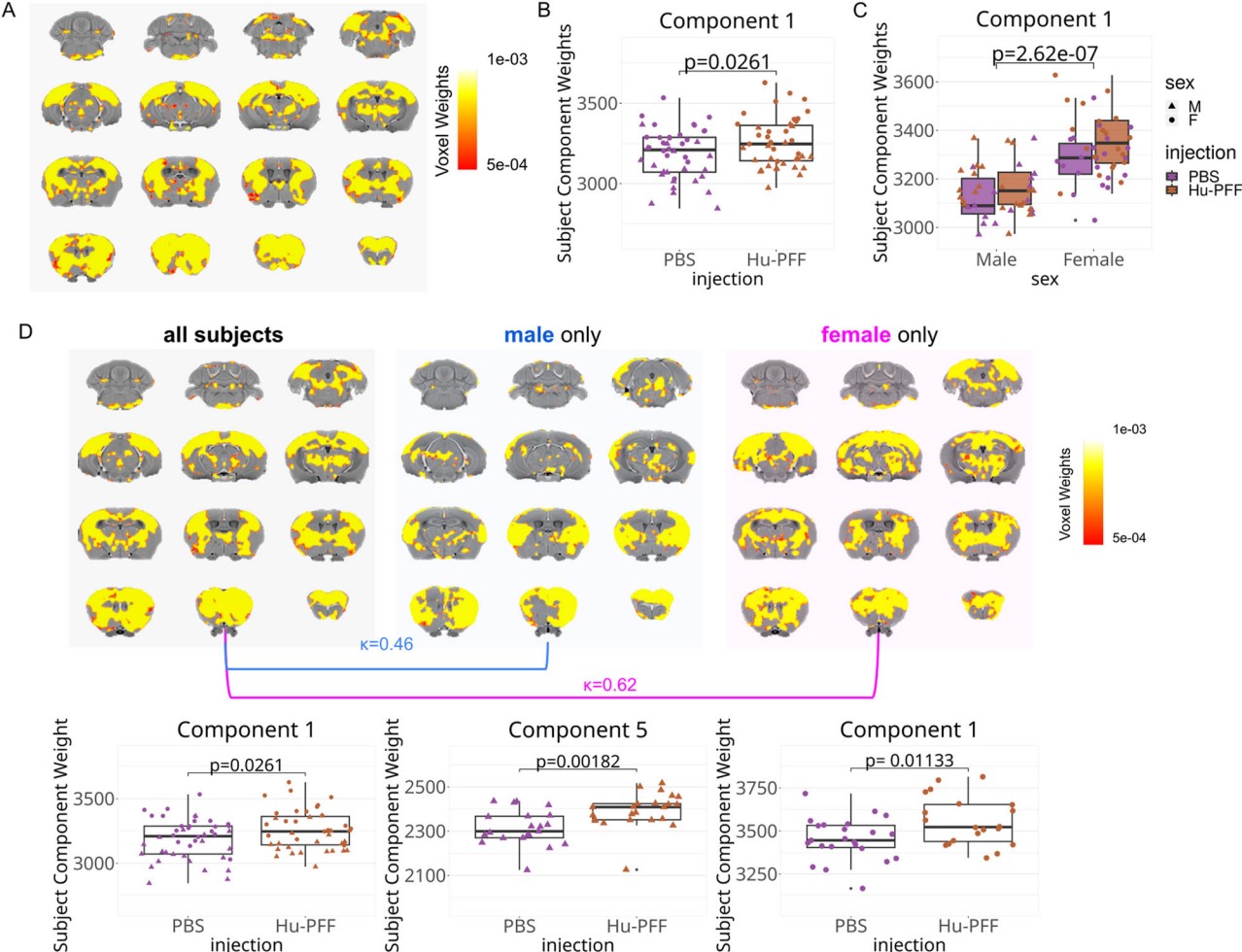

**Fig. 4 | Sex driven patterns of pathology at peri-motor symptom onset. A**, **D** The spatial pattern of voxel component weights (denoted by the color map) plotted onto the mouse brain, depicting patterns of voxels sharing a similar variance pattern. **A–D** OPNMF decomposition at 90 dpi (peri-symptom onset) for all subjects (*n* = 87 mice). Plot of subject weights shows significant **B** main effect of injection group (*p* = 0.0261; *d* = 2.266) and **C** main effect of sex (*p* = 2.62e−07; *d* = 5.621), but no significant interaction. Higher subject weights for Hu-PFF-injected mice (compared to PBS-injected nice) and for females (compared to males) for component 1 weights. **D** Sex specific OPNMF runs males only (*n* = 44 mice) (middle) versus females only (*n* = 43 mice) (right); *k* = 6 components, comparison versus all subject OPNMF run (left; shown as well in (**A**)). Significance for aSyn Hu-PFF-induced network level pathology assessed using general linear models; significant group differences in

subject component weights were observed for only one of six components for each of the three OPNMF runs (all subjects: component 1; male only: component 5 (*p* = 0.00182; *d* = 0.178); and female only: component 1 (*p* = 0.0113; *d* = 2.652)). Dice-Kappa overlay score was used to examine the spatial similarity between the three spatial patterns, such that a higher Kappa denotes greater overlap. The greatest overlap with the all subject OPNMF component is between it and the female-only spatial component (κ = 0.62), thus suggesting that the all subject component is predominantly female driven, despite observing no statistical group by sex interaction significance of the subject weights for the all subject component 1. Purple for PBS-injected mice, orange for Hu-PFF-injected mice, triangle points for male and circular points for female mice. The error bars indicate the data spread up to 1.5 times the interquartile range.

Analysis of the 120 dpi (once overt motor symptoms are present; $n = 32$) yielded components 3 and 6 which were the only two components where the subject weights differentiated the Hu-PFF from PBS injected groups, and whereby the Hu-PFF mice had higher weights, denoting patterns of PFF-induced spatial volumetric covariance (component 3: $p = 0.0359$; $d = 2.13$; component 6: $p = 0.0007$; $d = 3.88$). Unfortunately, we are underpowered to explicitly examine sex-specific covariance patterns given the high numbers of attrition due to symptom and disease progression (with <8 Hu-PFF male mice remaining). Nonetheless, we did observe a trend-level injection group by sex interaction ($p = 0.0769$; $d = 1.85$) for component 6 only, with slightly higher subject weights for male Hu-PFF-injected mice (Supplementary Fig. 9).

### Western blot

We next sought to examine whether there were inherent sex differences in the M83 model in the expression of the human aSyn transgene, as well as the normally expressed mouse aSyn that may represent a baseline difference rather than a difference in pathological accumulation. The levels of aSyn serine129 phosphorylation (pS129), a classical pathological modification of aSyn[72–75], did not differ between males and females when comparing three different brain regions (striatum, brainstem, and cortex) (Supplementary Figs. 10 and 11) ($n = 8$ mice; 4 males and 4 females). Similarly, human and total (human + mouse) aSyn did not differ between the sexes (Supplementary Fig. 10).

Additionally, we analyzed the RIPA-insoluble fraction of these same samples, since aSyn aggregates can be found in this fraction. The signal obtained for pS129 was weak (Supplementary Fig. 10), which is expected for healthy mice[16,76], and we found no sex differences for all variants of aSyn (pS129, human, and mouse) (Supplementary Fig. 10).

### Discussion

The findings of this study provide valuable insights into the complex interplay between biological sex, disease progression, and neuroanatomical alterations in synucleinopathies. Here, our investigations encompassed multiple aspects of disease pathogenesis and aSyn spreading over time, including longitudinal neuroanatomical alterations, structural covariance patterns of atrophy, and motor symptomatology assessment.

Neuropathological examination of the brain at different stages of PD progression has been extensively documented, and the revealed patterns of pathology heavily suggest aSyn spreading between anatomically connected regions over time in what have become classic studies of this phenotype[77]. Despite extensive examination of aSyn spreading in animal models of synucleinopathy[15,25,78–80] via widespread Lewy body-like deposition in the brain, affirmative evidence of atrophy being caused by this hypothesized pathogenic process is still lacking. Here, we examined longitudinal within subject measures of pathology over the disease time course in the same model of synucleinopathy from pathology seeded at a known central locus across the whole brain. Our findings agree with our previous cross-sectional study[24] that demonstrated that aSyn mediated neurotoxicity preferentially impacted regions highly interconnected with the injection site displayed the most severe pathology, such as basal ganglia areas and connected motor cortices. Most importantly however, in this study we are better suited to examine spreading and progressive pathology, and assess normative variations in disease progression, both in terms of brain pathology and motor symptomatology. Our results highlight the significance of sex-specific differences in the context of aSyn spreading and neurodegenerative disease modeling by presenting sex-specific spatial and temporal aspects of synucleinopathy-associated disease progression.

We observed more aggressive neurodegeneration (atrophy) in male Hu-PFF-injected mice. Moreover, the identification of sex-specific patterns of structural covariance at both peri-motor symptom onset and post-motor symptom onset time points underscores the need for a nuanced approach when studying disease progression and may suggest multiple sex-specific mechanisms at play. While both male and female mice displayed Hu-PFF-induced patterns of atrophy, we observed early widespread pathology in the

female mice that seems to be less toxic given that these mice survived longer than their male counterparts. However, once the mice showed overt motor symptomatology, the male Hu-PFF-injected mice displayed widespread pathology which coincided with their lower rates of survival (with higher rates of male Hu-PFF-injected mice reaching their humane endpoint prior to the experimental endpoint). The levels of host aSyn (mouse or human) and baseline phosphorylated S129 aSyn (the substrate for aSyn spreading and toxicity) were not different between sexes. Hence, it is likely that intrinsic sex-specific molecular, cellular or circuit differences in aSyn spreading and toxicity contribute to the observed differences in brain atrophy and survival. Whether these are hormonally regulated remains to be defined. This study represents a critical step forward in understanding the impact of sex on disease progression in preclinical M83 Hu-PFF mouse models of synucleinopathies. Our findings emphasize the importance of considering sex-specific pathologies for investigating disease mechanisms and therapeutic interventions.

Our findings not only underscore the importance of considering sex (and gender for human studies) differences in synucleinopathies but also reveal a gap in the existing literature. The scarcity of studies that systematically focus on such differences in these disorders is evident, despite reported and anecdotal phenotypic differences between men and women diagnosed with synucleinopathies. Overall, men are more likely than women to experience a higher prevalence, increased incidence, greater disease severity, and heightened susceptibility to synucleinopathies[26–30]. In terms of neuroanatomical changes between the sexes, unfortunately there is a lack of neuroimaging studies centered on sex differences in synucleinopathies and in the vast PD literature in general, in spite of the overwhelming clinical and epidemiological evidence supporting sex differences in diagnosis, presentation, and prevalence[81]. Namely, three structural MRI studies have explored sex differences in gray matter brain atrophy within the context of PD. In studies of cortical thickness, one study by Tremblay et al.[82] in de novo patients with PD did not observe any sex differences, whereas work by Oltra et al.[83] observed significant cortical thinning in various regions across the brain (including all four lobes: frontal, parietal, temporal, and occipital lobes) in male compared to female patients, similar to work by Yadav et al.[84] in patients undergoing treatment for PD. With regards to whole brain volume measures, using DBM, all three studies reported greater male atrophy in patients. Specifically, they observed more cortical regions with male-driven (eleven) versus female-driven atrophy (six), and greater subcortical atrophy in males[82–84]. Nonetheless, more evidence is needed to effectively parse through sex-specific differences in neuroanatomical pathology to be able to use neuroimaging techniques as valuable biomarkers for stratifying patients based on their risk, rate and severity of disease progression[85].

The predominant notion driving these sex differences is thought to be the role of hormones and sex chromosomes[28,86,87]. However, there is a lack of evidence for the associations between events of hormonal fluctuations in women (such as menstruation onset age, menopause onset age, fertile lifespan, pregnancy history, use of oral contraceptives and hormone replacement therapy) and the risk of developing PD in women[88–90]. Thus, there is still a lack of causality for the neuroprotective properties of estrogen[91]. Several biological, genetic, and hormonal factors have been proposed to explain these sex-specific differences.

One key hypothesis is the influence of hormonal differences. Estrogen, for instance, is known to exert neuroprotective effects through various mechanisms, such as reducing oxidative stress, inhibiting apoptosis, and promoting synaptic plasticity. Studies have shown that estrogen can modulate the expression of neurotrophic factors and reduce the production of inflammatory cytokines, which may delay the onset of motor symptoms[34]. Conversely, testosterone has been implicated in modulating neuroinflammation by exacerbating inflammatory responses and accelerating disease progression in males[34]. In addition to hormonal factors, immune system differences may play a role. Microglial cells, the brain's resident immune cells, exhibit distinct activation profiles between males and females. Female microglia might display a more regulated and less detrimental

response to neurodegenerative stimuli, while male microglia may be more prone to excessive activation, contributing to faster disease progression[92]. This difference in immune response could influence the extent and timing of neurodegeneration across sexes. Furthermore, synaptic and neuronal differences could contribute to the observed disparities. Females may exhibit greater synaptic plasticity and resilience to neurodegenerative changes, potentially due to the effects of estrogen on synaptic connectivity and neuronal survival[93]. Meanwhile, certain neuronal populations, such as dopaminergic neurons in the substantia nigra, might be more vulnerable in one sex compared to the other, leading to variations in the disease course. Lastly, metabolic differences in brain function, such as glucose utilization and mitochondrial efficiency, might also impact the progression of neurodegenerative diseases in a sex-specific manner[94]. These factors, along with the hormonal and immune mechanisms mentioned, suggest a complex interplay that requires further investigation to fully understand the biological underpinnings of sex differences in neurodegenerative diseases.

In fact, as stated in a recent systematic review on sex differences in movement disorders by Raheel et al.[34], several important questions remain unanswered with regards to the estrogen conversation: disentangling endogenous versus exogenous estrogen exposure, the causality of estrogen effects on different aspects of the disease, and the intricate interplay between hormonal changes and the progression of synucleinopathies, determining the threshold, time window (e.g., premenopause versus perimenopause versus postmenopause), and other potential modifying factors that influence the neuroprotective effect of estrogen[34]. Future studies should aim to address these questions, as well as explore the role of testosterone, immune responses, and other biological factors to provide a more comprehensive understanding of the mechanisms driving these sex differences.

Despite the insights provided by this study, it is essential to acknowledge its limitations. First, this study primarily focused on structural changes, and future research should investigate functional and molecular aspects of sex-specific differences in synucleinopathies. Previous work by our group investigated cellular markers of pathology (pS129syn, astrocytes and microglia) in relation to the MRI-derived brain pathology and observed that atrophied areas in the Hu-PFF-injected mice had elevated levels of these cellular markers (compared to PBS-injected mice) whereas the areas not involved in the MRI findings similarly did not have significantly different levels of these markers between the two injection groups[24]. Additionally, the regions implicated in the MRI analysis particularly had higher levels of astrocytes and microglia suggesting the compensatory role of these helper cells coming to the rescue prior to the elevated levels of pathogenic alpha-synuclein later observed[24]. However, how these findings relate to sex differences in the presentation of atrophy patterns and the underlying cellular pathology has yet to be elucidated. Nonetheless, this manuscript provides a much-needed characterization of the neuroanatomical pathological changes over the disease time course in vivo, similarly to human synucleinopathy imaging studies. Beyond this characterization, our specific focus on both sexes and sex differences allows us to examine sex-specific disease presentation and progression otherwise uncharacterized in this well-used model. Our findings advocate strongly for a shift in pre-clinical research practices to consistently include both male and female model systems to better model more than half the world's population. By acknowledging and addressing sex differences with investigation in both female and male mice, we aimed to enhance the translatability of these models, improving its face and construct validity. Nonetheless, further research is warranted to elucidate the underlying mechanisms driving these sex-specific variations, as the identification of the molecular pathways driving these differences may open new avenues for sex-specific therapeutic strategies, often neglected in many research avenues. We sought to examine basal differences in alpha synuclein expression in the M83 model between the sexes, while no significant findings were observed (though the study may have been underpowered to detect such subtle effects), these results suggest that the sex differences identified in this study are more likely related to variations in disease propagation, clearance, and inflammation, as discussed previously. Similarly,

beyond the importance of using both female and male mice for preclinical testing, given that the age of onset of such neurodegenerative diseases occurs around peri- and post-menopause, it has become clear that our modeling of the female condition in mice should include the hormonal fluctuations occurring during these phases to most accurately disentangle the role of hormones in disease pathogenesis. We thereby urge such investigations to also be considered. This paradigm shift is vital to drive advances in the field, improve our understanding of sex-specific mechanisms in synucleinopathies, effective translation from animal models to human organisms, and to pave the way for the development of novel, sex-specific therapeutic strategies. Ultimately, such efforts will contribute to the establishment of personalized medicine approaches that consider the unique needs of all patients affected by these devastating neurodegenerative diseases.

A final, yet important, limitation is the investigation of overt motor symptomatology when observing the mice in their home cage. These symptomatologies have been extensively characterized in this model over the years[9,15,25]. However recent evidence has emerged suggesting cognitive impairment may precede the motor symptomatology as evidenced by touchscreen cognitive tasks performed at pre-motor symptom onset time points (~50–60 days post-injection)[16,24]. This investigation of nonmotor symptomatology is important with regards to the clinical presentations of synucleinopathies wherein nonmotor symptoms (autonomic, olfactory, etc.) present prior to diagnosis[1,3,95]. Interestingly, for one of our motor tasks, we observed a relatively low latency to fall during the rotarod test (albeit in line with Masuda-Suzukake et al.[19], however, lower compared to other studies: Luk et al.[18]; Macdonald et al.[96]; Masuda-Suzukake et al.[20]), and notably, we did not detect significant injection group differences in performance despite the presence of overt motor symptomatology for the Hu-PFF-injected mice when ambulating in their cages (after unblinding). This inconsistency suggests that the rotarod test may not be sensitive enough to consistently capture subtle performance deficits associated with disease progression in our model. This observation further underscores the potential value of incorporating nonmotor symptomatology assessments. Quantifying nonmotor impairments could provide a more comprehensive understanding of the disease's progression and the sensitivity of symptom detection[16,24]. By integrating nonmotor symptom assessments with traditional motor tests, we may enhance our ability to detect early changes in disease presentation, ultimately improving the characterization of the disease phenotype and the evaluation of therapeutic interventions.

In conclusion, this study provides a clear example of why it is important to incorporate sex-specific considerations into preclinical mouse models of synucleinopathies. Our findings provide compelling evidence of sex-specific differences in the spatiotemporal dynamics of synucleinopathy-associated pathology and disease progression, underscoring the importance of considering sex-specific factors in research and clinical practice. Future research should aim to investigate the molecular and cellular mechanisms driving sex differences in synucleinopathy pathogenesis, as it may offer significant implications for patient management and the development of targeted therapeutic interventions. Furthermore, further investigation into diverse hormonal models could better capture the nuanced impact of these factors on disease susceptibility and presentation. Embracing this paradigm shift will be essential for advancing our understanding of disease mechanisms and developing tailored therapeutic strategies, bringing us closer to personalized medicine approaches for patients affected by synucleinopathies and other neurodegenerative diseases.

## Data availability
All source data (demographic, behavioral and western blot raw data) are available in an online repository (ref. 97; https://doi.org/10.5281/zenodo.14655729). The following MRI data have been made publicly available and include the following: DBM template brain and mask, statistical maps from longitudinal voxel-wise analysis, and the spatial patterns for each of the components from each of the four OPNMF runs. Any other information is available upon request.

## Code availability

All our methods and techniques (preprocessing, DBM, OPNMF, etc.) used in this manuscript are open source and available at our Github (https://github.com/CoBrALab). R scripts were also made publicly available (ref. 97; https://doi.org/10.5281/zenodo.14655729).

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

## Acknowledgements

S.T. was awarded a Healthy Brains, Healthy Lives fellowship and was awarded a Fonds de Recherche du Québec en Santé (FRQS) doctoral training scholarship. E.A.F. is supported by a CIHR Foundation grant (FDN–154301) and by a Canada Research Chair 1315 (Tier 1) in Parkinson's disease. M.A.M.P. and V.F.P. received support from the Canadian Institutes of Health Research, Natural Science and Engineering Research Council of Canada, CFI/ORF, The Tanenbaum Open Science Institute (TOSI), a BrainsCAN/Healthy Brains, Healthy Lives (HBHL; Canada First Research Excellence Fund to Western/McGill University) Initiative for Translational Neuroscience Award, and BrainsCAN Research Excellence Award Awards, TRanslational Initiative to DE-risk NeuroTherapeutics (TRIDENT) as well as support for the Rodent Cognitive and Innovation Core for behavior experiments. M.A.M.P. is a Tier I Canada Research Chair in Neurochemistry of Dementia. M.M.C. receives support from NSERC, CIHR, and BrainsCAN/HBHL Initiative for Translational Neuroscience Award, TRIDENT, and salary support from the FRQS (Chercheur Boursier Junior 2) and James McGill.

## Author contributions

S.T. with the help of E.D.C.P. and M.M.C. conceived, designed, and planned the experiments. S.T. along with J.S.H.P., D.G., M.P., and K.M. collected the MRI and behavioral data. W.L., E.D.C.P., and I.S. produced and characterized the PFF inoculum, under the supervision of T.M.D. and E.A.F. S.T. processed and analyzed the MRI and behavioral data. MRI preprocessing and DBM pipeline processing scripts were created by G.A.D. OPNMF interpretation was performed with the help of R.P. V.N. and R.S.C., under the supervision of M.A.M.P. and V.F.P., acquired and analyzed the western blot data. S.T. wrote the manuscript with support from M.M.C. All authors provided feedback on the manuscript.

## Competing interests

The authors declare no competing interests.
