## [Transparent Peer Review file · Communications Biology]

Female mice exhibit resistance to disease progression despite early pathology in a transgenic mouse model inoculated with alpha-synuclein fibrils

Corresponding Author: Ms Stephanie Tullo

Version 0:

Reviewer comments:

Reviewer #1

(Remarks to the Author)

This manuscript attempts to examine potential sex-mediated differences in the disease progression in a Het MM83 mouse model following alpha-synuclein injection by examining motor function, multimodal MRI and Western blot examination in a longitudinal manner. Identifying sex-mediated differences in disease progression is an important component of research that should be the focus of more studies.

The paper is generally well written with the introduction incorporating the key topics that the manuscript will set out to examine. Similarly, the methods also provide sufficient information for future researchers to follow.

Unfortunately the results section is very hard to follow as the majority of the figure panels are out of order relative to the text. For example, the results jump from Survival and Motor symptoms onset to rotarod then wire hang followed by pole test. The figure itself is the opposite.

I find some the graphs difficult to interpret. For example, the pole test. The mouse is placed on the pole and then must rotate 180 degrees and descend within a 2 min time frame. You state in the supplement materials that if the mouse falls the time would be recorded upon landing on the floor whereas failure to turn around and descend would result in 2min being recorded. Was falling considered as a failure of adequate motor function or lack of grip strength? The portion successful for the 120 dpi appeared to be relatively stable after 30 seconds for both groups. I would suggest including the average time taken to complete the task.

The same should be considered for the wire hang test.

Even though you state that no difference was observed between males and females, I believe that this data could be included in the supplementary data.

I'm surprised that the PBS injected mice performed so poorly in the rotarod. From the supplementary data it would suggest that, on average, all mice failed at around the 120 second mark (or 12 rpm) which is exceptionally low. can the authors offer and explanation to this?

Figure 3A states differences in M83 HuPFF over time however, the way the figure is set out is confusing as the first panel (left to right) are the representative sections of the brain and not the results. I would suggest even boxes around each of the data sets to make it easier to examine.

Once again, the results in the manuscript are not in order relative to the figure itself with some panels not being referred to at all.

There also does not appear to be the appropriate statistics included in the figure or figure caption, for example the rate of decline in the CP for figure 3N is hard to decipher.

Figure 4B appears to require the second panel to be either a separate panel or referred to as an insert for 4B to make it clearer what this finding is showing.

If the Western blots are an important component of the study why are they only a supplementary figure?

Often the data for both males and females is collapsed but not always justified in text. Even if there are no significant differences, it would be good to see the raw data in the supplementary figures.

Reviewer #2

(Remarks to the Author)

Tullo et al. have investigated the effect of sex on alpha-synuclein-induced neurodegeneration in the M83 mouse model using longitudinal magnetic resonance imaging. They observed that the male mice injected with alpha-synuclein reached humane endpoint earlier than the female mice as they developed aggressive pathology compared to female mice. However, there was no difference in onset of motor symptoms between the male and female mice. In general, both incidence and prevalence of Parkinson's disease (PD) is greater in men than in women, with men and women presenting distinctive motor and non-motor symptoms [1]. Although the study itself was executed well, the study design should have included behavioral testing for assessing non-motor symptoms as many transgenic rodent models that are currently used to understand PD show cognitive decline, anxiety- and depression-like symptoms [2]. A few of my concerns are listed below:

1. As brain imaging was one the main readouts, apart from biochemical analysis it would be interesting to see if the pathology (measured via immunohistochemical analysis) followed the brain atrophy. In many cases, the extent of pathology or in some cases even neuroinflammatory phenotype may correlate with behavioral measures.
2. As mentioned above, addition of other behavioral tests to measure cognition could have strengthened the study. At the minimum, the reason for not investigating non-motor symptoms should be included in the discussion.
3. At least n=9 mice survived till the endpoint, so why were the western blots run on n=3-4 per group, please explain if those numbers were sufficient with power analysis.
4. Statistical results should be completely reported with F values along with p values. It is actually easier if all the raw data were provided along with outliers (clearly indicated) and all the statistical results for all the figures.

References

1. Cerri, S., L. Mus, and F. Blandini, Parkinson's Disease in Women and Men: What's the Difference? J Parkinsons Dis, 2019. 9(3): p. 501-515.
2. Khan, E., I. Hasan, and M.E. Haque, Parkinson's Disease: Exploring Different Animal Model Systems. Int J Mol Sci, 2023. 24(10).

Reviewer #3

(Remarks to the Author)

This paper is an important study aiming at characterising the sex-specific differences of disease progression in a mouse model of Parkinson's disease. To achieve this goal, the authors combined behavioural testing and MRI morphometry over 4 months. As a preclinical MRI expert, I will mostly focus on the MRI methods and analysis.

First of all, I would like to emphasise the care taken by the authors in selecting state-of-the-art methods for both the MRI analyses and statistics. In addition to the already excellent level of this article, here are a few suggestions to further refine the paper.

1. In this study, the authors used PCA to compare global patterns of cerebral atrophy between sexes. While I appreciate the relevance and rigour along which this metric was used, I feel disappointed that the interpretation of the results remains within a very mathematical standpoint, and raises concerns that a less expert public would be lost and misled without proper explanations, e.g. identification of anatomical regions should be made more accessible if relevant.
2. While this is already covered by the authors in their discussion, the scope of this paper does not extend much further from signalling differences, which calls for further investigations. Mechanisms are hovered in the discussion, and digging deeper into their understanding would move this article's message beyond 'sex is important'.
3. In Figure 3, the authors identified several regions showing different regression patterns. Are these regions related to the observed behavioural differences?

Version 1:

Reviewer comments:

Reviewer #1

(Remarks to the Author)

The authors have undertaken an extensive review of the manuscript and addressed the major concerns raised. As a result, the clarity of the manuscript and figures have been much improved. I am now satisfied that this manuscript is suitable for publication. The authors should be congratulated for the care taken in revising the manuscript.

Reviewer #3

(Remarks to the Author)

The authors have addressed all my concerns.

We thank the editors and reviewers for their comments and suggestions. We have made significant edits to the manuscript that we believe improves the clarity of the manuscript and its scientific quality. We hope that the reviewers find that the submitted revision meets the high standards of Communications Biology. In the following, we provide our responses in italicized font, including the changes made to the manuscript quoted, and the comment from each reviewer is shown in bolded text.

Reviewers' comments:

Reviewer #1 (Remarks to the Author):

This manuscript attempts to examine potential sex-mediated differences in the disease progression in a Het MM83 mouse model following alpha-synuclein injection by examining motor function, multimodal MRI and Western blot examination in a longitudinal manner. Identifying sex-mediated differences in disease progression is an important component of research that should be the focus of more studies. The paper is generally well written with the introduction incorporating the key topics that the manuscript will set out to examine. Similarly, the methods also provide sufficient information for future researchers to follow.

We thank the reviewer for the extremely positive appraisal of our manuscript.

Unfortunately the results section is very hard to follow as the majority of the figure panels are out of order relative to the text. For example, the results jump from Survival and Motor symptoms onset to rotarod then wire hang followed by pole test. The figure itself is the opposite.

We apologize for this oversight. We are now mindful of the figure layout in relation to the presentation of the results in the text. We have made the corresponding edits to ensure a clearer and sequential layout throughout the manuscript.

I find some the graphs difficult to interpret. For example, the pole test. The mouse is placed on the pole and then must rotate 180 degrees and descend within a 2 min time frame. You state in the supplement materials that if the mouse falls the time would be recorded upon landing on the floor whereas failure to turn around and descend would result in 2min being recorded. Was falling considered as a failure of adequate motor function or lack of grip strength? The portion successful for the 120 dpi appeared to be relatively stable after 30 seconds for both groups. I would suggest including the average time taken to complete the task.

The same should be considered for the wire hang test.

We used two measures from both the pole and wire hang tests; specifically, the time taken and whether the task was performed successfully. To improve clarity, we have added the following to the supplementary method sections 7.6.1. Pole test and 7.6.2. Wire Hang.

7.6.1. Pole test

“The time required for the mouse to descend to the floor was recorded with a maximum duration of 2 minutes (120 seconds). Each subject completed three trials with a 30-minute rest between each trial, following the protocol detailed by Matsuura et al. (1997). Notably, two scores were recorded: 1) whether the task was performed successfully or was a failed attempt, and 2) the time taken to descend. If a mouse was not able to turn downward and remained at the top of the pole, the trial was marked as a failure and the maximum time allotted was noted (120 seconds). In cases where the mouse fell part of the way down but subsequently descended the rest, the behaviour was scored as a failure however the time it took to reach the floor was still noted. If the mouse fell for more than half the length of the pole, the behaviour was scored as a failure and the maximum time allotted was noted. For cases where a mouse fell immediately after placement at the top of the pole, a failed attempt, the trial was repeated.”

Having two scores for each of these tests allows for a better understanding of how the behaviour was performed. For instance, when a short duration was recorded for the pole test, the addition of a pass or fail score allows for the interpretation of whether the task was performed with or without impairments. However, in the most extreme case where the mouse could not descend and fell the length of the pole, a failed attempt, the maximum time allotted was noted.

For the wire hang test, the opposite applies:

7.6.3. Wire Hang

“Similar to the pole test, performance on the wire hang was examined in terms of latency and failure/success rates. If the mouse holds on for more than 3 minutes, it receives the maximum allotted time (180 seconds) and a successful score. For this task, there are three possible outcomes: 1) the mouse holds on for more than 3 minutes, 2) the mouse holds on for less than 3 minutes and receives a failure score with the duration recorded, or in the most extreme case, 3) the mouse is unable to hold on for at least 10 seconds. In the latter case, three attempts were allowed during the trial before a failure score and a duration of less than 10 seconds was recorded (of highest duration of the failed attempts).”

Given the nature of these tasks and their scoring, where performance was examined in terms of latency and failure/success rates, the best statistical approach is using a Cox Proportional Hazard model (Kassambara et al., 2018). This model is well-suited for analyzing time-to-event data, allowing us to account for both the time it takes for a task to be completed and the likelihood of failure or success. By incorporating these variables, the Cox model provides a more nuanced understanding of the factors influencing performance outcomes.

To improve the interpretability of the model and the figure, we added an explanation of how to read the figure for the pole and wire hang test in the figure caption (Figure 2).

[F,G,H,I] When interpreting the plot, the y-axis (Proportion Failed/Successful) shows the percentage of mice (or subjects) that failed to last more than 3 minutes on the wire hang test or successfully completed the pole test. The closer the line is to 100%, the more successful/unsuccessful the subjects were. The x-axis (Time in seconds) indicates the time taken to successfully complete the pole test/the amount of time they lasted on the wire under the 3-minute successful cut-off. As time increases, the plot tracks the proportion of subjects who were successful/failed by that time. Next, the lines show the cumulative success/failure rates over time for the two injection groups.

Although the reviewer notes that the proportion of successful trials stabilizes after 30 seconds for both groups, we believe that requires further clarification. The PBS group (purple) does seem to stabilize close to 75% success after 30 seconds, with little to no further increase. While the Hu-PFF group (orange) shows a noticeable plateau after 30 seconds, it does not reach the same level of stability as the purple PBS group. Importantly, these data show that the PBS (purple) group shows a higher proportion of success in less time, while the Hu-PFF (orange) group has a lower and slower success rate. This suggests that the PBS group is more proficient in the task than the Hu-PFF group. We agree that including the average time taken to complete the task for both groups would provide valuable additional insight, and we will include this information in the revised manuscript.

Regarding the suggestion to include the average time taken to complete the tasks, we have opted not to present means for this particular dataset. The reason is that the data for both the pole and wire hang tests are constrained by ceiling or floor effects, which make the use of averages misleading. For instance, in the wire hang test, if many mice reach the 180-second cut-off, the mean would artificially suggest better performance across the group, even in the presence of variability in other subjects' performances. Similarly, in the pole test, the 120-second maximum for failures creates a skewed mean that doesn't reflect the range of performance between successful and unsuccessful trials. Instead, using the Cox Proportional Hazard model, this framework provides a more accurate and robust way to analyze and interpret this type of data. This method captures both the time taken and the success/failure rates, offering a nuanced picture of performance across groups. To further clarify the data distribution, we have added a distribution plot below, which better illustrates the variation in task completion times for both groups. The plots below highlight the ceiling effect in the wire hang test and the floor effect in the pole test, where many subjects either reached the maximum allotted time for wire hang or failed to complete the task in pole test. We have decided to add the two figures below to the Supplementary document as it may be useful to the readers.

Methods section, section 2.7. Statistical analyses, subsection 2.7.1. Disease progression and motor symptomatology analysis

“Distribution of data, highlighting a ceiling and floor effect for wire hang and pole test respectively, is available in the Supplementary Figures S2 and S3.”

Supplementary Figure S2. Data distribution for wire hang test highlights ceiling effect. This histogram highlights the spread of the data such that there is a ceiling effect in the wire hang test, where many subjects reached the maximum allotted time for wire hang and accordingly successfully completed the test, as any latency under the 180 seconds is deemed a failed attempt. Using the Cox Proportional Hazard model, this framework provides a more accurate and robust way to analyze and interpret this type of data. This method captures both the time taken and the success/failure rates, offering a nuanced picture of performance across groups. PBS-injected mice are in purple and Hu-PFF mice are in orange. Histogram binning was performed at 5.

Supplementary Figure S3. Data distribution for pole test highlights floor effect. This histogram highlights the spread of the data such that there is a floor effect in the pole test, where many subjects failed to perform the task and consequently were allotted the maximum time (120 seconds). Using the Cox Proportional Hazard model, this framework provides a more accurate and robust way to analyze and interpret this type of data. This method captures both the time taken and the success/failure rates, offering a nuanced picture of performance across groups. PBS-injected mice are in purple and Hu-PFF mice are in orange. Histogram binning was performed at 2.

Given the nature of these distributions, it is not possible to perform standard parametric statistical models, such as *t*-tests or ANOVAs, because the data do not conform to the appropriate assumptions (specifically that of a normal distribution), which is clearly violated here. Instead, the Cox Proportional Hazard model is a more suitable approach for analyzing these data, as it is robust to censored outcomes and allows us to account for both task completion time and success/failure rates.

Even though you state that no difference was observed between males and females, I believe that this data could be included in the supplementary data.

We have now included these data in the Supplementary, as Supplementary Figure S5.

Supplementary Figure S5. Sex differences in motor symptomatology and performance on motor tasks. [A] Weight trend across disease progression. Significant inverted U-shaped trajectory for Hu-PFF-injected mice, with weight loss as of 90 dpi (compared to PBS-injected mice) ($p=0.0056$). No sex-specific differences were observed when examining the triple interaction between group, sex, and time (days post-injection). [B] Average rotarod performance across time showed no significant differences between injection groups and sex. [C,D] Wire-hang performance at 90 (left) and 120 (right) dpi. Significant difference in the proportion of mice that failed (<3 minutes) between injections groups ($p=0.0104$; $p=0.000504$), with higher rates of failure for the Hu-PFF-injected mice (red dashed line); no sex difference observed. [E,F] Pole Test performance at 90 (left) and 120 (right) dpi. Hu-PFF-injected mice had lower proportions of mice successfully passing the test, and took significantly longer to descend the pole compared to their saline injected counterparts ($p=0.0304$; $p=0.017$). Purple colour denotes PBS-injected mice and orange colour denotes Hu-PFF injected mice. Line type was used to denote each of the sexes: solid line (with blue shading) for male and dashed line (with red shading) for female mice, except when no sex differences are displayed solid lines then denote

both sexes grouped together. Data point shapes also denote the sex of the mice: triangle for males and round for females.

I'm surprised that the PBS injected mice performed so poorly in the rotarod. From the supplementary data it would suggest that, on average, all mice failed at around the 120 second mark (or 12 rpm) which is exceptionally low. can the authors offer and explanation to this?

We appreciate the reviewer's comment and understand the concern. While the reviewer is correct that all the mice failed at around the 120-second mark, this performance is consistent among the PBS control group at each of the 4 time points. We explicitly tested if there was a practice effect across the time points to determine if further training was needed to improve performance. However, we do not observe any such effect. The mice were properly trained, habituated to the behavioural room conditions prior to the test administration (1 hour habitation), and the mice were handled three times the week prior to the experimental time point to reduce anxiety and allow for optimal performance. In our study, the focus was on the relative difference between the PBS and PFF groups, rather than absolute performance. However, we acknowledge that the observed latency to fall might appear lower than typical rotarod performance in other studies. To address this concern, we will be adding this discussion point in the Discussion section regarding the training protocols, the observed performance levels, and the implications of the latency times in relation to other studies using similar methodologies.

Discussion section, last paragraph

"[...] Interestingly, for one of our motor tasks, we observed a relatively low latency to fall during the rotarod test (albeit in line with Masuda-Suzukake et al., 2013, however, lower compared to other studies: Luk et al., 2012b; Macdonald et al., 2021; Masuda-Suzukake et al., 2014), and notably, we did not detect significant injection group differences in performance despite the presence of overt motor symptomatology for the Hu-PFF-injected mice when ambulating in their cages (after unblinding). This inconsistency suggests that the rotarod test may not be sensitive enough to consistently capture subtle performance deficits associated with disease progression in our model. This observation further underscores the potential value of incorporating nonmotor symptomatology assessments. Quantifying nonmotor impairments could provide a more comprehensive understanding of the disease's progression and the sensitivity of symptom detection (Lackie et al., 2022; Tullo et al., 2023). By integrating nonmotor symptom assessments with traditional motor tests, we may enhance our ability to detect early changes in disease presentation, ultimately improving the characterization of the disease phenotype and the evaluation of therapeutic interventions."

Figure 3A states differences in M83 HuPFF over time however, the way the figure is set out is confusing as the first panel (left to right) are the representative sections of the brain and not the results. I would suggest even boxes around each of the data sets to make it easier to examine.

Thank you for the suggestion to improve the clarity of our figure; we have made the suggested edits to improve the data visualization (see below).

Once again, the results in the manuscript are not in order relative to the figure itself with some panels not being referred to at all.

As mentioned above, we are now mindful of the figure layout in relation to the presentation of the results in the text, and have made the corresponding edits to ensure a clearer layout throughout the manuscript.

There also does not appear to be the appropriate statistics included in the figure or figure caption, for example the rate of decline in the CP for figure 3N is hard to decipher.

The statistical models used for the results represented in Figure 3 are all linear models assessing differences in volume trajectories over time (days post-injection). However the variable of interest was different for each spatial representation (Figure 3A-D). The first analysis assessed the interaction of the injection group and time with sex as a covariate; we begin by demonstrating that there are group differences in voxel volumes over time between M83 Hu-PFF compared to M83 PBS mice, as shown in Figure 3A (and voxel trajectory plots E-H). The figures are arranged in a similar manner for this and all remaining comparisons. Next we assessed the impact of the PFF inoculation on each sex (two-way interaction of injection group and time; males in figure 3B and females in figure 3C; plots I-L). Finally, we assessed significant differences in voxel-wise volume trajectories between male and female Hu-PFF injected (Figure 3D; plots M-P).

Accordingly for the plots in E-L we chose to highlight voxels within regions where group differences over time were observed then examined if these effects would be observed in each sex-specific analysis. Our observations for sex differences are supported by this style of examination. In these regions we observed significant group differences over time for males but not females. Separately, plots M-P are displaying voxel-wise trajectories for the Hu-PFF mice only, now comparing the trajectories between male and female mice. Here, we picked peak voxels whereby we observed significant differences in trajectories; regions including the PAG, contralateral caudoputamen, ipsilateral primary motor cortex and anterior olfactory nucleus.

We have added this additional explanation to the figure caption of Figure 3 to improve the clarity of the reader.

“Figure 3. Sex differences in Hu-PFF-induced brain atrophy examined using voxel-wise volumetric trajectories over time. [A-D] Coronal slices of the mouse brain (from posterior to anterior) with t-statistical map overlay; demonstrating [A] the effects of injection over time in M83 hemizygous mice, [B] the effects of injection over time in male mice, [C] the effects of injection over time in female mice, [D] the effect of sex over time for Hu-PFF-injected mice. Colour map describes the direction of the t-statistics; cooler colours denoting negative values; most commonly corresponding to volume decline and warmer colours denoting positive values, corresponding to volume increases over time for the group of interest indicated. [E-P] Plot of relative volume change (mm^3) over the four time points (-7, 30, 90 and 120 days post-injection). We chose to highlight voxels within regions where group differences over time were observed (as observed in [A]) then examined if these effects would be observed in each sex-specific analysis

(as observed in [B] for the males and [C] for the females); peak voxel in [E,I] the right midbrain reticular nucleus (MRN), [F,J] right substantia nigra (SN), [F] the injection site (right caudoputamen (CP)), [H,L] right primary motor area (1 MC). Our observations for sex differences are supported by this style of examination. In these regions we observed significant group differences over time for males but not females. Purple line for PBS-injected mice, orange line for Hu-PFF-injected mice, solid line and triangle points for male and dashed line and circular points for female mice. Separately, plots M-P are displaying sex differences in voxel-wise trajectories for the Hu-PFF mice only, comparing the trajectories between male and female mice. Here, we selected peak voxels whereby we observed significant differences in trajectories; regions including the [M] PAG, [N] contralateral caudoputamen, [O] ipsilateral primary motor cortex and [P] anterior olfactory nucleus; orange points and line describe these Hu-PFF injected mice, where solid line, triangle points, and blue shading describes the male while the dashed line, circular points and red shading describe the trajectory for the female mice. Overall, volumetric decline was observed for Hu-PFF injected mice (compared to PBS-injected mice), with steepest rate of decline for male Hu-PFF-injected mice.”

Figure 4B appears to require the second panel to be either a separate panel or referred to as an insert for 4B to make it clearer what this finding is showing.

To improve the clarity of the figure, we have decided to split the two plots labeled in figure 4B into separate panels (4B and 4C).

If the Western blots are an important component of the study why are they only a supplementary figure?

We certainly believe that the Western Blots data was important for the current study and was used to verify if the observed sex differences were a product of basal differences in synuclein concentration or burden. Since the results were negative, we were able to rule this out. However, given that the results do not identify basal sex differences, we decided to include these results in the supplementary section.

Often the data for both males and females is collapsed but not always justified in text. Even if there are no significant differences, it would be good to see the raw data in the supplementary figures.

As mentioned above, we have now included these data in the Supplementary to further improve transparency of the data acquired.

Reviewer #2 (Remarks to the Author):

Tullo et al. have investigated the effect of sex on alpha-synuclein-induced neurodegeneration in the M83 mouse model using longitudinal magnetic resonance

imaging. They observed that the male mice injected with alpha-synuclein reached humane endpoint earlier than the female mice as they developed aggressive pathology compared to female mice. However, there was no difference in onset of motor symptoms between the male and female mice. In general, both incidence and prevalence of Parkinson's disease (PD) is greater in men than in women, with men and women presenting distinctive motor and non-motor symptoms [1]. Although the study itself was executed well, the study design should have included behavioural testing for assessing non-motor symptoms as many transgenic rodent models that are currently used to understand PD show cognitive decline, anxiety- and depression-like symptoms [2]. A few of my concerns are listed below:

1. As brain imaging was one the main readouts, apart from biochemical analysis it would be interesting to see if the pathology (measured via immunohistochemical analysis) followed the brain atrophy. In many cases, the extent of pathology or in some cases even neuroinflammatory phenotype may correlate with behavioural measures.

Previous work from our group has investigated cellular markers of pathology (pS129syn, astrocytes and microglia) with regards to the MRI-derived brain pathology (Tullo et al., 2023 J Neurochem.) We have shown that areas that had smaller volumes for the Hu-PFF-injected mice had elevated levels of these cellular markers (compared to PBS-injected mice) whereas the areas not involved in the MRI findings similarly did not have significantly different levels of these markers between the two injection groups. Additionally, we started to observe a pattern whereby regions implicated in the MRI analysis particularly had higher levels of astrocytes and microglia suggesting the compensatory role of these helper cells coming to the rescue prior to the elevated levels of pathogenic alpha-synuclein later observed. We expanded on this point in the discussion section.

Discussion section, 8th paragraph

“Despite the insights provided by this study, it is essential to acknowledge its limitations. First, this study primarily focused on structural changes, and future research should investigate functional and molecular aspects of sex-specific differences in synucleinopathies. Previous work by our group investigated cellular markers of pathology (pS129syn, astrocytes and microglia) in relation to the MRI-derived brain pathology and observed that atrophied areas in the Hu-PFF-injected mice had elevated levels of these cellular markers (compared to PBS-injected mice) whereas the areas not involved in the MRI findings similarly did not have significantly different levels of these markers between the two injection groups (Tullo et al., 2023). Additionally, the regions implicated in the MRI analysis particularly had higher levels of astrocytes and microglia suggesting the compensatory role of these helper cells coming to the rescue prior to the elevated levels of pathogenic alpha-synuclein later observed (Tullo et al., 2023). However, how these findings relate to sex differences in the presentation of atrophy patterns and the underlying cellular pathology has yet to be elucidated.”

2. As mentioned above, addition of other behavioural tests to measure cognition could have strengthened the study. At the minimum, the reason for not investigating non-motor symptoms should be included in the discussion.

We have previously investigated cognition in this mouse model (Tullo et al., 2023 J Neurochem.) and observed cognitive impairments, specifically with regards to reduced cognitive flexibility (using touchscreen testing), prior to the onset of motor deficits. Given that no previous work has investigated sex differences in this well-characterized model, we sought first to examine these key questions that provide a useful baseline for future studies of alpha-synuclein propagation in this model. Our plans are, indeed, to further investigate differences in spreading based mouse genotype (A53T mutation), species of the fibril injected (mouse versus human derived PFF), and location of the inoculation to assess the impact on disease progression, survival and symptom profile of the mice (motor and nonmotor impairments). We agree that there is more work to be done here. But at the moment, we believe that publishing these baseline findings are critical to furthering what we consider to be fundamental work on sex differences in models of neurodegeneration.

In line with the previous reviewer's comment, we have added to the Discussion section to speak to use of motor and nonmotor testing.

Discussion section, last paragraph

“A final, yet important, limitation is the investigation of overt motor symptomatology when observing the mice in their home cage. These symptomatology have been extensively characterized in this model over the years (Bétemps et al., 2014; Froula et al., 2019; Luk et al., 2012a; Luk et al., 2012b; Masuda-Suzukake et al., 2013; Masuda-Suzukake et al., 2014; Mougnot et al., 2012; Sacino et al., 2014; Watts et al., 2013). However recent evidence has emerged suggesting cognitive impairment may precede the motor symptomatology as evidenced by touchscreen cognitive tasks performed at pre-motor symptom onset time points (~50-60 days post-injection) (Lackie et al., 2022; Tullo et al., 2023). This investigation of nonmotor symptomatology is important with regards to the clinical presentations of synucleinopathies wherein nonmotor symptoms (autonomic, olfactory, etc.) present prior to diagnosis (Boeve et al., 2003; Kao et al., 2009; Pagonabarraga & Kulisevsky, 2012; Sveinbjornsdottir, 2016). Interestingly, for one of our motor tasks, we observed a relatively low latency to fall during the rotarod test (albeit in line with Masuda-Suzukake et al., 2013, however, lower compared to other studies: Luk et al., 2012b; Macdonald et al., 2021; Masuda-Suzukake et al., 2014), and notably, we did not detect significant injection group differences in performance despite the presence of overt motor symptomatology for the Hu-PFF-injected mice when ambulating in their cages (after unblinding). This inconsistency suggests that the rotarod test may not be sensitive enough to consistently capture subtle performance deficits associated with disease progression in our model. This observation further underscores the potential value of incorporating nonmotor symptomatology assessments. Quantifying nonmotor impairments could provide a more

comprehensive understanding of the disease's progression and the sensitivity of symptom detection (Lackie et al., 2022; Tullo et al., 2023). By integrating nonmotor symptom assessments with traditional motor tests, we may enhance our ability to detect early changes in disease presentation, ultimately improving the characterization of the disease phenotype and the evaluation of therapeutic interventions.”

3. At least n=9 mice survived till the endpoint, so why were the western blots run on n=3-4 per group, please explain if those numbers were sufficient with power analysis.

The western blot experiments were conducted using mouse brains derived from non-injected M83 transgenic mice in order to assess whether baseline differences in alpha-synuclein expression were present between the male and female M83 mice. Our goal was to seek basal differences in the M83 model, which we did not observe. Therefore, the sex differences are likely to have more to do with differences in propagation, clearance, and inflammation (as we have discussed). We agree that more work could be done in later time points, and will seek to do this important work in subsequent studies. Nonetheless, we conducted a quick group power analysis based on our current means and standard deviations for the RIPA striatum total alpha-synuclein; the sample size per sex needed to detect an effect size with 80% power at a 0.05 significance level is ~ 34 subjects, which is unmanageably high.

Discussion section, 8th paragraph

“[...] Nonetheless, further research is warranted to elucidate the underlying mechanisms driving these sex-specific variations, as the identification of the molecular pathways driving these differences may open new avenues for sex-specific therapeutic strategies, often neglected in many research avenues. We sought to examine basal differences in alpha synuclein expression in the M83 model between the sexes, while no significant findings were observed (though the study may have been underpowered to detect such subtle effects), these results suggest that the sex differences identified in this study are more likely related to variations in disease propagation, clearance, and inflammation, as discussed previously. [...]”

4. Statistical results should be completely reported with F values along with p values. It is actually easier if all the raw data were provided along with outliers (clearly indicated) and all the statistical results for all the figures.

With the expectation to fully report our statistical results, we chose to present both p-values and effect sizes in our findings. While p-values indicate statistical significance, effect sizes provide a clearer understanding of the strength of the observed effect, making the results more comprehensive, meaningful, and readily usable for downstream meta-analyses. We recognize that simply reporting significance is insufficient, which is why we aimed to offer a fuller picture of the data's impact. Moreover, in the spirit of transparency and open science, we included all raw data by plotting every data point in our figures, without excluding any "outliers" or applying

different treatments to any data point. We are unclear on why the reviewer asserts that we removed outliers. No outliers were detected or removed in this manuscript.

References

1. Cerri, S., L. Mus, and F. Blandini, Parkinson's Disease in Women and Men: What's the Difference? *J Parkinsons Dis*, 2019. 9(3): p. 501-515.
2. Khan, E., I. Hasan, and M.E. Haque, Parkinson's Disease: Exploring Different Animal Model Systems. *Int J Mol Sci*, 2023. 24(10).

Reviewer #3 (Remarks to the Author):

This paper is an important study aiming at characterising the sex-specific differences of disease progression in a mouse model of Parkinson's disease. To achieve this goal, the authors combined behavioural testing and MRI morphometry over 4 months. As a preclinical MRI expert, I will mostly focus on the MRI methods and analysis.

First of all, I would like to emphasise the care taken by the authors in selecting state-of-the-art methods for both the MRI analyses and statistics. In addition to the already excellent level of this article, here are a few suggestions to further refine the paper.

1. In this study, the authors used PCA to compare global patterns of cerebral atrophy between sexes. While I appreciate the relevance and rigour along which this metric was used, I feel disappointed that the interpretation of the results remains within a very mathematical standpoint, and raises concerns that a less expert public would be lost and misled without proper explanations, e.g. identification of anatomical regions should be made more accessible if relevant.

We thank the reviewer for their feedback and for highlighting the need to make our results more accessible to a broader audience. We apologize for any confusion caused, and we would like to clarify that we used orthogonal projective Non-Negative Matrix Factorization (OPNMF), not Principal Component Analysis (PCA), to examine spatial covariance patterns of atrophy, as has been used previously in this model (Tullo et al., 2023). To address the reviewer's concerns regarding interpretability, we sought to provide more accessible explanations of the NMF results, highlighting the anatomical regions associated with each identified pattern of atrophy. We will also ensure that technical jargon is minimized or clearly defined to make the content more comprehensible for non-expert readers.

Results section, 3.3. Whole brain structural covariance patterns of atrophy

“In addition to examining volumetric trajectories of decline to toxicity derived from Hu-PFF injection, we sought to elucidate if this decline was occurring in a network-like pattern. To examine this, we used a multivariate statistical methodology similar to methods that have previously been used in clinical PD (Zeighami et al., 2015) and that are consistent with our

previous work (Tullo et al., 2023). The technique we have favoured, namely the orthogonal projective non-negative matrix factorization (OPNMF) (Patel et al., 2020; Robert et al., 2022; Sotiras et al., 2015; Yang & Oja, 2010), uses the voxel-wise measures of volumetric differences as input and outputs. It provides voxel-level spatial components that covary together and subject-specific weights related to how much each subject loads onto a spatial pattern.”

3.3.1. Peri-motor symptom onset

Similar to the analysis performed in Tullo et al., (2023), at 90 dpi, we examined the spatial patterns for $k=6$ components (as determined by stability analysis; see section 7.8. Orthogonal projective non-negative matrix factorization in Supplementary Materials; Supplementary Figure S3). The focus of this analysis was to examine data-driven patterns of PFF-induced atrophy and their sex differences. General linear models for each of the 6 components revealed no significance with sex-by-injection group interaction that predicted subject weights subject-wise loading onto the spatial pattern (Figure 4A). One of the six components statistically separates the Hu-PFF- from PBS-injected groups (component 1; $p=0.0323$; $d=2.179$) (Figure 4B). The spatial covariance pattern of this component (component 1) consisted of voxels in regions with known connections to the injection site, such as voxels within striatal-pallidal-midbrain areas, as well as strong cortical and thalamic involvement (Figure 4A). All 6 components are detailed in Supplementary Figure S4. For this component (component 1), we observed a significant effect of sex ($p=2.18e-8$; $d=6.23$), with higher subject weights for all females (Figure 4C), suggesting that female mice (regardless of injection group) had stronger associations with this spatial covariance pattern than the males. We used these findings to support further examination of sex-specific structural covariance patterns.

For each sex-specific OPNMF, we similarly observed one of the six components with significantly higher subject weights for the Hu-PFF-injected mice, compared to their saline counterparts (component 5 for the male only analysis ($p=5e-5$; $d=0.2445$) and component 1 for the female only ($p=0.00963$; $d=2.716$) (see Supplementary Figure S5 and S6 for all 6 components in each sex-specific analysis).

Both the male- and female-specific Hu-PFF-induced patterns (component 5 and component 1 respectively) consisted of voxels within the same regions mentioned above for the all subjects OPNMF analysis; which included: voxels in the basal ganglia and thalamus, anterior to posterior dorsal cortical areas, and in the midbrain and pons (Figure 4D). To further examine the degree of similarity between each sex-specific OPNMF and the original all subject OPNMF, Dice-kappa overlap scores were used to examine the degree of overlap between each sex-specific and all-subjects patterns as a proxy for sex-specific contributions. Our analysis revealed a higher degree of overlap between the females and the all subjects pattern ($\kappa=0.62$) comparatively to the male only overlap ($\kappa=0.46$) (Figure 4D). These findings suggest that, in contrast with our univariate analysis that reveals more aggressive localized neurodegeneration in male mice, there is a more spatially widespread neurodegeneration pattern for the female mice compared to their male counterparts associated with the PFF injection.”

2. While this is already covered by the authors in their discussion, the scope of this paper does not extend much further from signalling differences, which calls for further investigations. Mechanisms are hovered in the discussion, and digging deeper into their understanding would move this article’s message beyond ‘sex is important’.

We appreciate the reviewer's feedback and agree that further investigation into the underlying mechanisms would significantly enhance the impact of this study. While our primary aim was to establish the existence of sex-specific differences in cerebral atrophy patterns using NMF, we recognize that a deeper exploration of the mechanisms driving these differences is crucial for a more comprehensive understanding.

In the current study, we focused on neuroanatomical differences as a first step to identify and quantify these potential sex-specific patterns of neurodegeneration. Although we have touched upon potential mechanisms in the discussion, we acknowledge that these are speculative and require further empirical evidence. We have expanded the discussion to include a more detailed consideration of the potential pathways and biological processes that could underlie the observed differences, and outline future research directions that could help elucidate these mechanisms more thoroughly.

Discussion section, fifth paragraph

"[...] Several biological, genetic, and hormonal factors have been proposed to explain these sex-specific differences.

One key hypothesis is the influence of hormonal differences. Estrogen, for instance, is known to exert neuroprotective effects through various mechanisms, such as reducing oxidative stress, inhibiting apoptosis, and promoting synaptic plasticity. Studies have shown that estrogen can modulate the expression of neurotrophic factors and reduce the production of inflammatory cytokines, which may delay the onset of motor symptoms (Raheel et al., 2023). Conversely, testosterone has been implicated in modulating neuroinflammation by exacerbating inflammatory responses and accelerating disease progression in males (Raheel et al., 2023). In addition to hormonal factors, immune system differences may play a role. Microglial cells, the brain's resident immune cells, exhibit distinct activation profiles between males and females. Female microglia might display a more regulated and less detrimental response to neurodegenerative stimuli, while male microglia may be more prone to excessive activation, contributing to faster disease progression (Bilal Tariq, Lee & McCullough, 2022). This difference in immune response could influence the extent and timing of neurodegeneration across sexes. Furthermore, synaptic and neuronal differences could contribute to the observed disparities. Females may exhibit greater synaptic plasticity and resilience to neurodegenerative changes, potentially due to the effects of estrogen on synaptic connectivity and neuronal survival (Uhl, Schmeisser & Schumann, 2022). Meanwhile, certain neuronal populations, such as dopaminergic neurons in the substantia nigra, might be more vulnerable in one sex compared to the other, leading to variations in the disease course. Lastly, metabolic differences in brain function, such as glucose utilization and mitochondrial efficiency, might also impact the progression of neurodegenerative diseases in a sex-specific manner. These factors, along with the hormonal and immune mechanisms mentioned, suggest a complex interplay that requires

further investigation to fully understand the biological underpinnings of sex differences in neurodegenerative diseases (Arioglu-Inan & Kayki-Mutlu, 2023).”

3. In Figure 3, the authors identified several regions showing different regression patterns. Are these regions related to the observed behavioural differences?

We thank the reviewer for their insightful comment. To provide a clearer link between the neuroanatomical and behavioural findings, we performed a simple correlation analysis between the eight regions identified in the regression patterns and the three behavioural tasks (rotarod, pole test, and wire hang) at the 120 dpi time point. However, we did not observe any significant correlations between these regions and the behavioural outcomes prior to multiple comparisons correction.

While this simplified analysis may be informative, it may not be suitable for capturing the complexity of the relationship between brain structure and behaviour in our model. Recent work by Marek et al. (2022) highlights the challenge of replicating brain-wide association studies (BWAS), examining the relationship between brain structure and behavioural phenotypes, using sample sizes appropriate for classical brain mapping (of ~25 subjects). When using exceptionally large datasets with ~ 50,000 subjects, the authors found that BWAS associations were smaller than previously thought, leading to inflated effect sizes and replication failures. Furthermore, given this work conducted with human data, its application to mouse work, where sample sizes are traditionally smaller, will undoubtedly be underpowered and not yield reliable results.

Given these constraints, any meaningful analysis of the relationships between brain regions and behavioural outcomes would require a much larger sample size and more sophisticated modeling techniques. Furthermore, investigating the longitudinal aspect of these relationships, i.e. how brain structure and behaviour evolve over time, would deservingly necessitate an entirely separate analysis, given its complexity. The incorporation of time points and the dynamics of brain-behaviour interactions adds another layer of analysis that would require careful consideration and may warrant its own publication.

Correlation between Brain Regions and Behavioral Measures

SNr: substantia nigra pars reticulata; SNc: substantia nigra pars compacta; PAG: periaqueductal gray; MRN: midbrain reticular nucleus; MOp: primary motor cortex; CP: caudoputamen; AON: anterior olfactory nucleus.